

# Modelling of atmospheric variability of gas and aerosols during the ACROSS campaign 2022 in the greater Paris area: evaluation of the meteorology, dynamics and chemistry

Ludovico Di Antonio[1], Matthias Beekmann[2], Guillaume Siour[1], Vincent Michoud[2], Christopher Cantrell[1], Astrid Bauville[1], Antonin Bergé[2,a], Mathieu Cazaunau[1], Servanne Chevaillier[1], Manuela Cirtog[1], Joel F. de Brito[3], Paola Formenti[2], Cecile Gaimoz[1], Olivier Garret[4], Aline Gratien[2], Valérie Gros[5], Martial Haeffelin[6], Lelia N. Hawkins[7], Simone Kotthaus[8], Gael Noyalet[1], Diana Pereira[2], Jean-Eudes Petit[5], Eva Drew Pronovost[7], Véronique Riffault[3], Chenjie Yu[2], Gilles Foret[1], Jean-François Doussin[1], Claudia Di Biagio[2]

[1]Univ Paris Est Creteil and Université Paris Cité, CNRS, LISA, F−94010 Créteil, France
[2]Université Paris Cité and Univ Paris Est Creteil, CNRS, LISA, F−75013 Paris, France
[3]Centre for Energy and Environment, IMT Nord Europe, Institut Mines-Télécom, Université de Lille, Lille, 59000, France
[4]Ville de Paris, Service parisien de santé environnementale, 75013, Paris, France
[5]Laboratoire des Sciences du Climat et de l'Environnement, CEA−CNRS−UVSQ, IPSL, Université Paris−Saclay, 91191 Gif−sur−Yvette, France
[6]Institut Pierre-Simon Laplace (IPSL), CNRS, École Polytechnique, Institut Polytechnique de Paris, 91128 Palaiseau CEDEX, France
[7]Department of Chemistry, Harvey Mudd College, 301 Platt Blvd, Claremont, California 91711, United States
[8]Laboratoire de Météorologie Dynamique (LMD-IPSL), CNRS, École Polytechnique, Institut Polytechnique de Paris, 91128 Palaiseau Cedex, France
[a]now at Laboratoire des Sciences du Climat et de l'Environnement, CEA−CNRS−UVSQ, IPSL, Université Paris−Saclay, 91191 Gif−sur−Yvette, France

*Correspondence to*: Ludovico Di Antonio (ludovico.diantonio@lisa.ipsl.fr), Matthias Beekmann (matthias.beekmann@lisa.ipsl.fr)

**Abstract.** The interaction of anthropogenic and biogenic emissions around large urban agglomerations remains an important question for atmospheric research and the key question of the ACROSS (Atmospheric Chemistry of the Suburban Forest) project. ACROSS is based on an intensive field campaign in the Paris area, including ground–based measurements in the urban inner center to suburban and forest sites, and on–board aircraft, during the exceptionally hot and dry summer 2022. 3D–modelling represents an important tool in ACROSS to disentangle processes such as emissions, transport and physico–chemical transformations. Here we use the available measurements from the ACROSS campaign in addition to observations from air quality and meteorological networks to evaluate the coupled WRF–CHIMERE model simulation. We find that the WRF model is able to reproduce the meteorological variability during the campaign, in particular two heat waves at the beginning and at the end. The model reproduces the daily ozone maxima well, but overestimates PM$_{2.5}$ by a factor of 1.5–2, partly due to an overestimation of secondary aerosol, both organic and inorganic. This overestimation was unexpected, and could be related to the specific hot summer conditions. For organic aerosol in the Ile–de–France area, the biases are reduced





to about ±20%. The model allows to explain how the interplay of different processes affects the fine aerosol variability and chemical composition over the campaign sites during two heatwave days: biogenic secondary organic aerosol formation in different forests around Paris, advection of wildfire aerosols, and long-range transport of Saharan dust.

## 1 Introduction

Megacities and large urban agglomerations are important hotspots of air pollution (e.g., Baklanov et al., 2010; Molina and Molina, 2004), where several key questions remain concerning their interaction with regional pollution. A first question concerns the share of pollution produced locally within urban areas and that imported from outside (Pandolfi et al., 2020; Lenschow et al., 2001). This aspect has been addressed for several megacities, such as Mexico City (Calderón-Garcidueñas et al., 2015), New York (Sun et al., 2011), London (Harrison et al., 2012), and Paris (Beekmann et al., 2015), evidencing a generally unexpectedly large regional contribution to urban pollution. For instance, Beekmann et al. (2015) show that regional inflow contributes up to 70% of the fine particulate matter ($PM_{2.5}$) concentrations for the Paris agglomeration. In addition, a recent analysis of long–term aerosol optical depth (AOD) measurements from the high–resolution MAIAC (Multi–Angle Implementation of Atmospheric Correction) product over 21 large European cities shows that, although most of the aerosol load comes from regional sources, urban emissions contribute to the degradation of air quality by increasing AOD levels by 57, 55, 39, and 32% on average for large metropolitan agglomerations such as Barcelona, Lisbon, Paris and Athens, respectively (Di Antonio et al., 2023).

A second important question concerns the impact and fate of concentrated urban emissions on regional pollution levels. From airborne observations in the plume of the Paris megacity during the MEGAPOLI campaign, Freney et al. (2014) showed that anthropogenic secondary organic aerosol (ASOA) accumulates as air masses move away from the urban agglomeration. A still debated question is the interaction between anthropogenic emissions, typically from the urban agglomeration, and biogenic emissions, from within, but also from surrounding forested areas, which are typical of megacities. Both types of emissions tend to enhance secondary pollutants (e.g. ozone, particulate matter), and it is important to quantify the contribution of each source. A large number of interactions (i.e., of additional effects beyond simple mixing) have been theoretically predicted, observed and modelled. For instance, both anthropogenic and biogenic species affect the oxidant capacity of the atmosphere ($OH$, $NO_3$, $Cl$, $O_3$), which determines the production of secondary pollutants from anthropogenic and biogenic precursors (Sartelet et al., 2012; Shrivastava et al., 2019). The yield of biogenic secondary organic aerosol (BSOA) formation can be both increased or decreased in the presence of anthropogenic emissions (Liu et al., 2021). For instance, under a set of specific anthropogenic–biogenic conditions, namely low NO levels, in presence of acidic aerosols, often involving sulfate, isoprene forms BSOA with high yield (e.g. (Carlton et al., 2018)).

The interaction of anthropogenic and biogenic emissions in and around a large urban agglomeration is the central scientific question of the French and international ACROSS project (Atmospheric Chemistry of the Suburban Forest, https://across.aeris-data.fr/, last access: 24 April 2024) (Cantrell and Michoud, 2022), focussing on the Paris agglomeration. With about 12 million



inhabitants, the Paris agglomeration is one of the largest megacities in Europe. It is surrounded by flat terrain including several large forested domains, thus an ideal playground to study anthropogenic-biogenic emission interactions. Some key questions of the ACROSS project are: how do anthropogenic–biogenic interactions impact primary pollutant oxidation pathways and secondary pollutant formation? How do they affect the organic carbon, reactive nitrogen and radical budgets? How these interactions affect the optical and hygroscopic aerosol properties, and, by consequence, their radiative effects? A large multi–site and multi–platform field campaign held from June 15th to July 25th 2022 has been the core of the ACROSS project. The campaign deployment consisted of three main instrumented super–sites on a NE–SW transect going from the Paris center to a sub–urban and a forested site, and several secondary sites with less complete instrumentation to study specific processes or to characterise the spatial variability of major pollutants. The French ATR 42 research aircraft was deployed to document the evolution of the Paris pollution plume at the regional scale. Summer 2022 over Western Europe was exceptionally hot and dry and is considered as a proxy for future climate conditions (Ribes et al., 2022). Indeed, the ACROSS campaign was characterised by two strong heat waves towards the beginning and the end of the measurements period, and a more temperate period in–between. These meteorological conditions varying between extremes, together with two episodes of transport of African dust over Europe and the high occurrence of fires during the period (Menut et al., 2023), made the campaign especially interesting to study also in the perspective of climate change and its impact on atmospheric composition and air quality.

Regional 3–D simulation with the WRF–CHIMERE model system (Menut et al., 2021) were performed covering the ACROSS field campaign domain and the whole of France for the June–July 2022 period, in order to support interpretation of campaign data: from local to regional scales or focussing on specific processes. In this paper, we present the configuration of the WRF–CHIMERE model simulation and first analysis of its output dataset. Based on several examples, we derive the spatio–temporal variability of pollutants during the ACROSS campaign using a combination of model data and observations. In order to assess the robustness of the model output for its use in subsequent studies, and with particular emphasis at the challenging conditions during the ACROSS period, a thorough model evaluation is presented. This evaluation relies on the already available in situ measurements from the ACROSS campaign in the Ile–de–France area, but also on long–term in situ measurements from established networks. A particular focus is placed on the evaluation of the meteorological simulations by the WRF–CHIMERE modelling system. This work shows how different transport patterns and meteorological conditions, especially during the two heat waves, affected regional concentrations of major pollutants such as ozone and $PM_{2.5}$, through the interaction of anthropogenic and biogenic emissions.

## 2 Methods

### 2.1 WRF–CHIMERE model configuration

The WRF–CHIMERE model (Menut et al, 2021, https://www.lmd.polytechnique.fr/chimere/, last access: 3 July 2024) is a 3–D regional Eulerian chemistry–transport model (CTM), which has been applied both for research and forecasting purposes over France and over many other countries (Cholakian et al., 2018; Ferreyra et al., 2016; Lachatre et al., 2020; Tuccella et al.,



2019). In this work we adopted the WRF version 3.7.1 (Skamarock et al., 2008) and CHIMERE v2020r3 coupled version (WRF–CHIMERE). Three one–way nested domains with spatial resolutions of 30, 6 and 2 km have been considered (Fig. 1). The 30 km domain covers the European continent and extends up to the Sahara Desert in order to be able to catch Saharan dust transport events over Europe. The 6 km domain covers the metropolitan France, while the 2 km domain covers the Paris

area and the Ile–de–France region where the ACROSS field campaign was conducted. The simulation extends from the June 15th to July 25th 2022, with two weeks of model spin–up. The 30 and 6 km domains have 15 vertical layers between the surface and 300 hPa, while 10 levels are used for the 2 km domain up to 500 hPa. In order to run the CHIMERE model, the meteorological data, initial and boundary conditions for the chemistry, anthropogenic, biogenic and fire emissions and land use have to be provided. All the data sources adopted for this work are summarised in Table 1. The WRF v 3.7.1 coupled with

the CHIMERE model provides the meteorological data online (i.e. meteorological fields are exchanged at every physical time step of ten minutes with the CTM for the three domains). In this study, the National Centers for Environmental Predictions (NCEP) analysis with 1°x1° spatial resolution has been used to constrain the meteorological initial and boundary conditions. The microphysics follow the Thompson scheme (Thompson et al., 2008), while the Yonsey University (YSU) planetary boundary layer (PBL) scheme was adopted for the PBLH estimation (Hong, 2010), that utilises the thermodynamic layer

detection based on the bulk Richardson number. Radiation fluxes are calculated using the Rapid Radiative Transfer model for General Circulation Models (GCMs) (RRTMG, Iacono et al., 2008). The nudging (i.e. the simulated meteorology relaxed towards the analysis) has been activated outside the PBL every six hours of simulation.

Gas and aerosols initial and boundary conditions to the WRF–CHIMERE model are taken from the CAMS global reanalysis product at 0.75°x0.75° of spatial resolution. Anthropogenic emissions are provided by the CAMS–GLOB–ANT v5.3 products

with a 0.1°x0.1° spatial resolution for 17 activity sectors (Soulie et al., 2023) and then reported to the 11 SNAP (Selected Nomenclature for reporting of Air Pollutants) sectors. For each activity sector, a chemical volatile organic compound (VOC) profile is used from Passant et al. (2002), and VOC are then regrouped into those present in the SAPRC chemical mechanism used. The gas–phase chemistry mechanism adopted for the ACROSS campaign is SAPRC–07A, a reduced scheme based on the SAPRC mechanism from Carter (2010). The biogenic emissions are simulated with the Model of Emissions of Gases and

Aerosols from Nature (MEGAN) v.2.1 module (Guenther et al., 2006) implemented within the CHIMERE model. Biogenic emissions depend on several parameters: the short–wave radiation, the surface temperature simulated by the WRF model and the leaf area index (LAI) derived from MODIS observations (Yuan et al., 2011). In this study we use the LAI referring to the 2013 year with a temporal resolution of 8–days and the emission factors from Sindelarova et al (2014). The emission scheme of biogenic VOCs considers a dependence on temperature and radiation, but not on soil humidity, and so the possible effects

of dryness. Fire emissions are taken from the CAMS Global Fire Assimilation System (GFAS, Kaiser et al., 2012) and processed with the emiFIRES_CAMS v2020r1 pre–processor (provided by the CHIMERE developers, https://www.lmd.polytechnique.fr/chimere/, last access 03 July 2024). The land use is based on GLOBCOVER with a spatial resolution of about 300 m (Arino et al., 2008).





The CHIMERE aerosol module simulates the aerosol concentration through a sectional bin approach. In this work, ten bins

between the 0.01 and 40 µm diameter range have been employed. Six of the ten bins fall in the submicron fraction of the size distribution. The lower and upper cut off diameters of the nearest bin to 1 µm are respectively 0.5 and 1.1 µm. The aerosol simulated species include black carbon (assumed as elemental carbon), sulfate, nitrate, dust, sea salt, primary particulate matter (PPM), and different primary (POA) and secondary organic aerosol (SOA) species. SOA are simulated based on the volatility basis set (VBS) scheme (Donahue et al., 2006), allowing to take into account functionalization (transfer to lower volatility

bins) processes. The scheme was later extended to include fragmentation (transfer to higher volatility bins) and non–volatile aerosol formation (Shrivastava et al., 2013, 2015) and adapted by Cholakian et al. (2019) for use in CHIMERE. The intermediate–VOC (IVOC) and the semi–VOC (SVOC) from POA are partitioned into nine volatility bins according to their saturation concentration $C^*$ ranging from 0.01 to $10^6$ µg m$^{-3}$. The POA can be oxidized by OH to form the Oxidised POA (OPOA), with the possibility of forming more functionalized (O2POA, O3POA) and non–volatile (ONVSOA) products. Four

different volatility bins in the 1 to 1000 µg m$^{-3}$ saturation concentration $C^*$ range and a non–volatile species have been used to represent the ASOA and the BSOA from VOC oxidation by OH, $NO_3$ and $O_3$. Fragmentation occurs at a 75% rate independently of the subsequent volatility (Shrivastava et al., 2015), leaving 25% to functionalization. These percentages are based on the best agreement between simulated and measured SOA as described in Shrivastava et al., (2013). Non–volatile biogenic and anthropogenic SOA (BNVSOA, ANVSOA) are formed considering a reaction rate constant corresponding to a

1h tropospheric lifetime (Cholakian et al., 2018). The total number of aerosol species formed from anthropogenic VOC (AVOC) and biogenic ones (BVOC) is 10 (5 for ASOA and 5 for BSOA). This VBS aerosol scheme contains therefore 40 species, which, multiplied by 10 aerosol bins, lead to 400 simulated species for the SOA. Note that primary fire organic aerosol is not included in the VBS scheme, but it has been considered as chemically inert. Nucleation, coagulation, condensation and dry and wet deposition processes are also addressed within this aerosol module.

**2.2 Datasets for model evaluation**

Datasets available to discuss the campaign period analysis and for model evaluation integrate ACROSS field campaign ground–based observations and larger–scale databases. In addition to ACROSS several projects contribute to the multi–project initiative PANAME with a centralised data base developed by AERIS (https://paname.aeris-data.fr/, last access, 3 July 2024), such as the ACTRIS EU Green deal project RI–URBANS (https://riurbans.eu/, last access, 3 July 2024).

In situ surface observations of the aerosol refractory and non–refractory chemical composition in the PM$_1$ fraction and aerosol–based mixed layer height (MLH; see Kotthaus et al. 2023 for discussion of ABL sublayer definition) were retrieved at the three ground–based sites of the ACROSS campaign:

(i)     the Paris–Rive Gauche (PRG) site (48.8277° N, 2.3806 °E), hosted at the Lamarck building at Université Paris Cité, on the south–eastern part of the Paris administrative borders, in a dense urban environment; instruments

were installed on the 7$^{th}$ floor of the building.



    (ii)     the SIRTA (Site Instrumental de Recherche par Télédétection Atmosphérique, 48.7090° N, 2.1488° E), located around 20 km south–west of the Paris administrative borders, is a suburban site due to its lower population density in an environment mixing forest, urban areas as well as agricultural fields and traffic roads (Bedoya-Velásquez et al., 2019; Zhang et al., 2019; Chahine et al., 2018; Haeffelin et al., 2005);

(iii)    the Rambouillet forest supersite (48.6866° N, 1.7045° E; hereafter RambForest), is located around 50 km south–west of the Paris administrative borders, in the middle of the Rambouillet national forest, a mixed deciduous and evergreen trees. Measurements have been performed at the ground level, within the forest canopy, as part as the operations of the Portable Gas and Aerosol Sampling Units (PEGASUS) mobile facility but also at the top of a 40 m high tower, above the canopy which is about 25 m high.

Data from the three ACROSS measurement sites include:

-    Submicron ($PM_1$) aerosol non–refractory composition (organic, nitrate, sulfate, ammonium, chloride) measured either by a Time–of–Flight Aerosol Chemical Speciation Monitor (ToF–ACSM, Aerodyne Research Inc., (Fröhlich et al., 2013), 6−min resolution) at the PRG and SIRTA sites, or by a High–Resolution TOF Aerosol Mass Spectrometer (HR–TOF–AMS, Aerodyne Research Inc., ((DeCarlo et al., 2006), 3−min resolution) at RambForest (Di Biagio et al., 2024; Ferreira de Brito et al., 2023b, a). The uncertainty on the total non−refractory mass concentration from ACSM is evaluated to be around 25% (Budisulistiorini et al., 2014; Crenn et al., 2015).

-    Equivalent black carbon (eBC) concentrations from dual–spot aethalometer (AE33, Magee Sci., 1–min resolution) (Di Antonio et al., 2023b), were processed by applying a site–invariant multiple scattering coefficient $C_{ref}$ of 2.45 as recommended by the pan–European ACTRIS (Aerosol, Clouds and Trace Gases Research Infrastructure) programme (ACTRIS, 2023; Savadkoohi et al., 2023).

-    Elemental and organic carbon (EC, OC) concentrations in the $PM_1$ fraction obtained by thermo–optical analysis on quartz filter (Pallflex Tissuquartz) samples collected over daytime (from 4h to 20h UTC) and night–time (from 20 h to 4h UTC) at PRG and RambForest (Pereira et al., 2024), using a continuous high volume aerosol sampler DHA–80 (DIGITEL enviro–sense, Switzerland).and analyzed following the EUSAAR–2 protocol (Cavalli et al., 2010). Organic matter (OM) was calculated assuming an OM–to–OC ratio of 1.8 (Sciare et al., 2011).

-    Refractory black carbon (rBC) concentrations at RambForest from a single particle soot photometer (SP2, DMT, 1 min resolution) (Yu and Formenti, 2023).

-    Mixed layer height (MLH) derived automatically from profile observations obtained by a network of automatic lidars and ceilometers (ALC) operated in synergy with the PANAME initiative (Kotthaus et al., 2023). At SIRTA ALC profile data (Lufft CHM15k) are processed using the STRATfinder algorithm (Kotthaus et al., 2020), and subjected to additional quality control that was developed in the context of RI–URBANS and the ABL testbed program (https://ablh.aeris-data.fr/, last access: 3 July 2024).

-    $PM_1$, $PM_{2.5}$, and $PM_{10}$ aerosol concentrations at PRG from FIDAS 200E at 1 min resolution (Di Antonio et al., 2023c).



In addition to the surface observations of the ACROSS sites, the aerosol submicron non–refractory composition and eBC
hourly data from ACSM and AE33 measurements over whole France have been extracted from the GEOD'AIR (GEstion des
données d'Observation de la qualité de l'AIR) database (https://www.geodair.fr/, last access: 3 July 2024), which is fed by the
regional air quality monitoring networks. Measurements from 9 sites were considered (Table S1). The AE33 eBC data were
treated assuming a $C_{ref}$ of 2.45 (ACTRIS, 2023; Savadkoohi et al., 2023).


Hourly surface level observations of $PM_{2.5}$, $PM_{10}$, $NO_2$, and $O_3$ across Europe from the European Environmental Agency
(EEA) database (https://discomap.eea.europa.eu/map/fme/AirQualityExport.html, last access: 24 April 2024) were further
considered to assess aerosol and ozone concentrations.

Finally, the Met Office Integrated Data Archive System (MIDAS) (Met Office, 2012)), which includes Météo France surface
observations, was considered for surface hourly meteorological parameters (temperature, pressure, relative humidity, wind
direction and speed) used to validate the surface meteorology of the WRF–CHIMERE simulations.

## 2.3 HYSPLIT backtrajectories

HYSPLIT (Stein et al., 2015) back–trajectory simulations at the ACROSS ground–based sites (PRG, SIRTA and RambForest)
were performed for the entire field campaign period using the WRF–CHIMERE meteorological fields as input using the
domains shown in Fig. 1 (Siour and Di Antonio, 2023). The back–trajectories have a time resolution of ten minutes, which
represents the exchange time between the WRF and CHIMERE models due to the coupling. More details on the product are
available at https://across.aeris-data.fr/ (last access: 3 July 2022).

## 3 Description of the meteorological situation and evaluation

**3.1 Evolution of the meteorological situation during the ACROSS campaign – between heat waves and oceanic flux**

Here we give a first overview of the different meteorological conditions during the ACROSS campaign period based on in situ
meteorological measurements at the suburban SIRTA site and aerosol measurements at the urban background PRG site, and
also at a broader scale from meteorological observations and analysis data over Western Europe. Hourly–averaged air
temperatures at the suburban SIRTA site (Fig. 2b) allowed a broad classification into three periods: (1) a first heatwave period
from June 15 to 18 with daily temperature maxima ($T_{max}$) between nearly 30°C and more than 36 °C; (2) a second long–lasting
colder period from June 19 to July 11 with $T_{max}$ between 15 °C and 25 °C; (3) a second heatwave period from July 12 to 23
with a mixture of very hot, but also colder days, with $T_{max}$ varying between 22 °C and 38 °C. The relative humidity (RH) is
strongly anti–correlated with temperature as expected, and varied from as low as 20–30% during the hottest days to more than
90% during the colder nights. EUMETSAT satellite images available at (https://www.wetterzentrale.de, last access 11 April



2024), show mainly clear sky conditions during June 15 to 18 and July 12 to 18, and mixed cloudy and clear sky conditions from June 19 to July 11 and from July 19 to 23, over the Ile–de–France region.

– **First heatwave period (June 15th – June 18th 2022)**

During the two hottest days of the first heatwave on June 17 and 18, moderate winds from the south–east (SE) (Fig. 2c) were measured at the SIRTA site. This corresponds to a generally southerly advection of hot and dry Mediterranean air masses and

possibly a Saharan origin linked to a cut–off low located west of the Iberian Peninsula and a ridge over France visible in the 500hPa geopotential map of June 18 12 UTC (Fig. 3a). Daily temperature maxima over France were mostly over 35 °C except for some mountainous or coastal areas, even reaching above 40°C in the south–west (SW) of France (Fig. 3o). This southerly advection is also well depicted in the 850 hPa temperature map (not shown) making evident that the French heatwave area was well connected to the Saharian heat reservoir. Linked to this is long range transport of Saharan dust to Northern France visible

through a large contribution to coarse dust aerosol (up to 40 µg m$^{-3}$, $PM_{10}$– $PM_{2.5}$ concentrations, Fig. 2a) at PRG on June 18 presumably indicative of coarse dust aerosol.

– **Clean period (June 19th – July 11th 2022)**

During the following colder and cleaner period, different sub–periods could be identified based on wind speed and direction variations: a first one from June 19 to 22, with moderate winds (1–3 m s$^{-1}$) coming from North (N) to NE bringing continental

air masses to Paris in addition to local pollution, resulting in still elevated $PM_{2.5}$ levels of 10 to 15 µg m$^{-3}$ nearly comparable to those during the first heat wave period. Then, from June 23 to July 2, south–westerly and westerly winds brought clean oceanic air masses to the sites and northern France. A typical 500 hPa geopotential field for this period (Fig. 3b) shows a low pressure system extending from Iceland to the British Isles and Benelux and a high pressure system over the Mediterranean Sea, resulting in the westerly flow. Daily temperature maxima were in the 15–20 and 20–25°C ranges over France (Fig. 3p).

A last period from July 3 to July 11, characterised again by mainly northerly and slightly stronger winds (3–4 m s$^{-1}$), led again to clean conditions ($PM_{2.5}$ at PRG from 3–7 µg m$^{-3}$).

– **Second heatwave (July 12th – July 25th 2022)**

The second heatwave period from July 12 onwards showed again varying meteorological conditions, but included four hot to very hot days with $T_{max}$ between 32 and 38 °C (Fig. 2b) and in general southerly winds of moderate speed (from 2 to 3 m s$^{-1}$,

Fig. 2c). For the hottest day on July 19, the 12 UTC 500 hPa geopotential field shows a cut–off low west of Britanny and a pronounced ridge extending northward up to southern Scandinavia (Fig. 3c). The 850 hPa maps show again a tongue of hot air over western and central Europe related to the Saharan heat reservoir (not shown). This ridge system is moving eastward, letting at its rear enter relatively colder air masses into south–western France with $T_{max}$ in the range of 25–35 °C, while $T_{max}$ is in the range of 35–40 °C over eastern France (Fig. 3q). The southward winds are again linked to Saharan dust transport for

July 13 and 19, indicated again by large coarse $PM_{10}$ of about 50 µg m$^{-3}$ at the PRG site (Fig. 2a). The large and dry conditions



over western France have induced strong forest fires (Menut et al., 2023) which were subsequently advected to the Paris region for several hours in the evening of July 19 and leading to enhanced $PM_{2.5}$ concentrations (> 60 µg m$^{-3}$ at the PRG site).

## 3.2 Evaluation of the WRF–CHIMERE meteorology

The ability of the WRF–CHIMERE model to simulate major meteorological variables was assessed for the following
parameters: the mean and maximum daily temperature, wind speed and direction, and mixing layer height (MHL). Daily maximum air temperature at 2 m above ground ($T_{max}$) is in general well simulated, with biases during the period below ±1 °C for the majority of the sites. This statement holds both for the entire campaign period and the three previously identified meteorological periods (Fig. 4, a–d). Larger biases (exceeding ± 2 °C) are encountered for mountainous areas along an axis from the eastern Pyrenees to the Alps, through the Massif Central, and the French Jura. Considering the average of all sites,
the mean bias error (MBE) for the whole period is –0.53 °C, which is slightly larger for the first and second heatwave periods (–0.74 °C and –0.64 °C, Fig. S3). Average correlation over the whole period and across all sites is 0.93 (Fig. S2). Daily maximum temperatures are especially well simulated for the high temperature periods, as can be seen for the 90$^{th}$ percentiles in Figure S4. For the mean daily temperature, differences between simulations and observations are in general smaller than those for $T_{max}$ (MBE = –0.32 °C, Fig. S3e).

Concerning the wind speed, a slight overestimation is generally observed for the ensemble of sites: biases are most often within the ±1 m s$^{-1}$ range for the different periods (Fig. 4, i–n). On the contrary, for the Ile–de–France region, a slight underestimation up to about –0.5 m s$^{-1}$ is noted for most of the sites. On the average of all sites in the model domain, a positive bias of +0.3 m s$^{-1}$ is observed when compared to the MIDAS sites (Fig. S3i). This overestimation is more pronounced for light wind days. For both wind speed and direction, the correlation is above 0.9 (Fig. S3).

The observed daily evolution of the MLH at the suburban SIRTA site is overall well reproduced by the planetary boundary layer height (PBLH) derived from WRF model fields, with daily maxima in the early afternoons, and minimum values at night (Fig. 5a). Correlations of hourly values are respectively 0.77, 0.92 and 0.88 for the three periods (Fig. 5b–d). Daily maxima are captured (± 200 m) for the majority of days, however, for about a third of cases the simulated PBLH remains below the observed MLH by more than 200 m. For two very hot days with $T_{max}$ higher than 33°C (June 18 and July 13), the observed
MLH at SIRTA nearly reached 3 km, while simulated PBLH are only 1600 and 1900 m, respectively. On June 18, it is generally challenging to define the MLH that is here derived using atmospheric aerosol as a tracer as on this day, the elevated layer of Saharan dust is entrained into the convective boundary layer during the day. The MLH observational product is hence associated with a certain uncertainty on that day which could explain the discrepancy to the PBLH that is determined from model data using a thermodynamic approach. On July 13 again, no elevated aerosol layer was present above the atmospheric
boundary layer and the observed MLH is very well defined. Further investigation is needed to understand the lower PBLH obtained from the model results. Given the MLH measurement product analysed here is mostly limited to layer height information above 230 m (Kotthaus et al. 2020), insights on the nocturnal stratification remain limited. The simulated PBLH frequently reports values below 200 m a.g.l. and as low as 20 m, i.e. the first possible level from WRF. These consistent night



time biases are more a result of the measurement product limitations and no conclusions can be drawn on the uncertainty of
the modelled nocturnal layer heights from this comparison.

## 4 Analysis and WRF–CHIMERE model evaluation for major pollutants during the ACROSS campaign

### 4.1 Analysis of regulated pollutants at the French scale

In this section, the broad features of the evolution of major pollutants, $O_3$, $NO_x$, $PM_{2.5}$ and $PM_{10}$, at the regional scale over
France during the campaign period are presented. We rely on observations from the French air quality networks (in particular
on sites representing background conditions), as examples of typical or significant days during the three periods (Fig. 3). First
we analysed the spatio–temporal evolution of the daily ozone maximum ($O_{3max}$) as a tracer of photo–chemical activity during
the three periods (Fig. 3 d–f). As expected, $O_{3max}$ is much larger during the heatwave days than during the clean period (Ancellet
et al., 2024). On June 18, the day with the largest $O_{3max}$ values during the first heatwave period, daily ozone maxima were in
the range of 108–144 µg m$^{-3}$ for western and central France including the Ile–de–France region. They were in the 144–180
µg m$^{-3}$ range in south–eastern France, where some sites exceeded the French pollution information threshold of 180 µg m$^{-3}$.
Nevertheless, this heatwave period did not correspond to a major ozone pollution episode in the Ile–de–France region and the
northern half of France, despite hot temperatures and low winds. HYSPLIT backward trajectories (arriving at 12 UTC at Paris
centre, 500 m height, Fig. S1, top) show that air masses had turned clockwise around the Ile–de–France region. The air masses
have then stayed over relatively low emission areas over rural France and the south of England the two previous days, which
may explain the relatively moderate ozone peaks.

During the following intermediate clean period, air masses were from oceanic origin, with daily ozone maxima below 108,
and even down to 72 µg m$^{-3}$ on July 1. During the second heatwave, on July 19, daily $O_3$ maxima showed a pronounced SW–
NE gradient. In northern France (including Ile–de–France) and south–eastern France, values were in the 144–180 µg m$^{-3}$ range,
while they remained below 108 µg m$^{-3}$ along the SW French coastline (Fig. 3e). This pattern was in good spatial correlation
with the $T_{max}$ field for this day (Fig. 3f). Indeed, the ozone heat pools moved eastwards, allowing clean oceanic air masses to
enter into SW France on their backs.

For particulate matter ($PM_{2.5}$ in Fig. 3g–i, $PM_{10}$ in Fig. 3l–n), concentrations were again larger during the heatwave periods.
On June 18, daily average $PM_{10}$ concentrations showed a pronounced E–W gradient, with values in the eastern part of the
country mostly in the 20 –30 µg m$^{-3}$ range, while they reached 30–40 µg m$^{-3}$ in the NW and 40–50 µg m$^{-3}$ in SW France.
Those elevated values are most likely due to dust transport from the Sahara, considering the enhanced simulated AOD for this
day over SW France and the Mediterranean region (Fig. S1, buttom), and are concomitant with heat advection from the same
region. As already mentioned, $PM_{10}$ peaks are also observed at the PRG urban site during the afternoon of June 18 (Fig. 2a).
HYSPLIT back trajectories for this day starting at 2 km height show air masses originating from Morocco three days before
(Fig. S1, top). The PM E–W gradient is still visible for $PM_{2.5}$, daily average concentrations being in the range of 0–14 µg m$^{-3}$



for most of the sites in eastern France, while in the range of 14–21 µg m$^{-3}$ for many sites in the western part, and even in the

21–28 µg m$^{-3}$ range for some sites in the Ile–de–France region.

During the intermediate clean period, on July 1, the PM$_{2.5}$ daily averages were below 14 µg m$^{-3}$ (Fig 3h) and the PM$_{10}$ below

20 µg m$^{-3}$ (Fig 3m).

During the second heat wave, the largest PM$_{2.5}$ daily averages above 35 µg m$^{-3}$ occurred at several sites in SW France values

(Fig 3i) due to the large fire activity in this region, caused by repeated dry and hot weather conditions. They were still enhanced

in Ile–de France and the NW of France, in the 14–21 and 21–28 µg m$^{-3}$ ranges, respectively, probably partly related to fire

plume transport to these regions (see discussion in Sect. 5.2). PM$_{10}$ daily averages show large values at the same sites in SW

France (PM$_{10}$ higher than 30 µg m$^{-3}$), probably again due to long range dust transport (Fig 3n).

## 4.2 Model evaluation for major pollutants at the French scale

This section presents the evaluation of the WRF–CHIMERE model simulations ability to reproduce the absolute concentrations

of ozone, NO$_2$, PM$_{2.5}$ and PM$_{10}$ and their spatio–temporal variability at the 6 km horizontal resolution.

The statistical analysis over the whole campaign period indicates that the daily maximum ozone concentrations are reasonably

well simulated by the WRF–CHIMERE model, which is able to represent its spatial and temporal variability. For most of the

French sites, negative O$_3$ biases fall in the 0–5 µg m$^{-3}$ range, while they can reach up to 20 µg m$^{-3}$during the first heat wave in

the Ile–de–France region (Fig. 6 a–d). They are also larger in mountain regions. For all sites, ozone shows an average mean

bias error (MBE) of –3 µg m$^{-3}$ corresponding to a normalized mean bias (NMB) of –4% (Fig. S5a). This figure shows that the

agreement is rather homogeneous for the different percentiles of ozone concentrations. For all the sites and daily ozone

maxima, the spatio–temporal correlation is 0.84 (Fig. 5a), with relatively small spatial heterogeneity (Fig. S4a).

For NO$_2$ daily means, average biases are between –5 to +5 µg m$^{-3}$. Negative biases are prevailing in the Ile–de–France region

(Figure 6e–h), maybe due to the fact that urban features are not well enough resolved in the simulation with a 6 km resolution.

The mean bias over all sites is –0.7 µg m$^3$ (Fig. S5e), mean NMB is –6%, and correlation is 0.51, but with a strong spatial

heterogeneity. As a matter of fact, sites in the Alpine regions in SE France or over the Massif Central mountains in central

France show close to zero or even negative correlations, indicating that the WRF–CHIMERE model does not capture well the

NO$_2$ variability for sites affected by orography (Fig S4e).

For PM$_{10}$, biases in average concentrations during the campaign are similar for most of the sites and close to zero or slightly

negative (down to –5 µg m$^{-3}$) (Fig. 6o). The mean bias (MBE) is negative (–1.3 µg m$^{-3}$) and corresponds to a NMB of –8%

(Fig. S5o). The negative bias is larger during the clean period (NMB; –17%), while it is only –5% and +2% during the first

and second heat waves (Fig. S5p and S5r). The fraction of comparisons with less than a factor of two difference is always

large, generally above 90%. Correlation is high (r = 0.8), with a rather homogeneous distribution among sites (0.6–0.9).

Conversely, the PM$_{2.5}$ fraction is overestimated in WRF–CHIMERE simulations. For most of the sites, this overestimation

falls within the 0–5 µg m$^{-3}$ range on average for the whole campaign (Fig. 6i), but it is even larger during the first heatwave

and in the Ile–de–France region (5–10 µg m$^{-3}$, Fig. 5l). The mean average positive bias is 6.0 µg m$^{-3}$ (MBE), which corresponds




to an NMB of +73% (Fig. S5i). The average temporal correlation is as large as 0.84, with a rather homogeneous distribution among the sites (0.7–0.95, Fig 4i), thus WRF–CHIMERE is able to reproduce the spatio–temporal $PM_{2.5}$ variability, despite this overestimation. Due to the slight underestimation of $PM_{10}$ and the pronounced overestimation of $PM_{2.5}$, the coarse PM ($PM_{10} - PM_{2.5}$) is underestimated, even if an explicit comparison was not performed.

## 4.3 Analysis and model evaluation of aerosol chemical composition at the French scale

The aerosol chemical composition is analysed to provide further insight into the submicron ($PM_1$) fraction. The analysis is based on ACSM and aethalometer (AE33) measurements performed at an approximate $PM_1$ cut–off sampling head employed within the French GEOD'AIR network (see Table S1) and compared to the simulated submicron aerosol composition. Figure 7 shows an overall broad agreement in the simulated and observed chemical composition. For most of the sites, the organic fraction makes up 60 –70% of the $PM_1$ mass, followed by sulfate (15–25%), nitrate, ammonium and black carbon (less than 10 % with the exception of nitrate for one site). These observations are compared to size bins integrated up to 1.1 μm in CHIMERE simulations.

Statistical evaluation presented in Fig. 8 shows a systematic overestimation of the organic fraction of (+21% in terms of NMB), sulfate (+16%), ammonium (+37%), nitrate (+53%), but only 7% for BC. For nitrate, the overestimation is most striking, a fraction of simulations shows nitrate concentrations in the 10 μg m$^{-3}$ range, as compared to observations of only a few μg m$^{-3}$. Reasons for this behaviour need to be further investigated. It could be due to poorly captured ammonium nitrate partitioning, or related to the fact that part of nitrate is captured in larger particles of terrigenic origin, and then not detected by ACSM, a process which is not included in the simulations. These overestimations are modulated by meteorological conditions. They are stronger for the organic fraction under heat waves conditions, and especially larger for peak concentrations, probably triggering excessive production of BSOA. On the contrary, for secondary inorganic aerosol, overestimations are stronger during the clean period (Fig. 8). Correlation is about 0.4 for all species except organic aerosol (r = 0.6). Overall, this analysis of different submicron species is coherent with the previously made evident $PM_{2.5}$ overestimation, although the extent of overestimation is smaller. Thus, aerosol species other than those analysed here (non–refractory species and BC), such as dust or primary mineral particles, are probably overestimated to an even stronger extent. Unfortunately, no measurements for these species were available during the campaign period.

## 4.4 Analysis and model evaluation of aerosol chemical composition at the ACROSS sites in the greater Paris area

The ACSM and AMS derived $PM_1$ non–refractive aerosol chemical composition measurements for PRG (Fig.S6), SIRTA (Fig.S7), and RambForest (Fig.S8) and the corresponding simulations at 2 km horizontal resolution, show many similarities. Across all sites, organic aerosol is the most abundant species, henceforth driving the $PM_1$ variability with enhanced concentrations during the two heat wave periods. For most days, sulfate is the second most abundant species, while pronounced nitrate peaks appear on specific days.



Contrary to the French GEOD'AIR sites, organic aerosol is not anymore systematically overestimated by simulations at the three ACROSS sites (Fig. 9). Reasons for this behaviour are not clear and need to be further investigated. For the whole set of available measurements, relative biases (in terms of NMB) are –20% for PRG, –3% for SIRTA and +21% for RambForest (Tables 2, S2, S3 and S4). From Fig. 9 and Fig. S6–8, a tendency for overestimations for $PM_1$ chemical species is noted during heatwave periods. Conversely, during the clean period, relative biases are negative in general. For instance, for SIRTA, the

NMB is +23% and +9% for the first and second heatwaves, while it is –25% for the clean period (Table S3). Similar tendencies are observed for the other two sites (Tables S2 and S4). Stronger positive biases during the heatwave periods have already been noted in the previous section for the GEODAIR data set.

Several simulated strong organic aerosol (OA) peaks show only weaker signatures in the observations. For instance, OA peaks

in the night from morning from June 17 to 18 exceeding 40 µg m$^{-3}$ are simulated at RambForest and SIRTA, and of approximately 30 µg m$^{-3}$ at PRG. The observed peak is only up to 25 µg m$^{-3}$ at RambForest, while observations are missing at SIRTA, and concentrations are about 15 µg m$^{-3}$ at PRG with a rather flat diurnal variation. These enhanced values are due to the advection of a plume of large BSOA concentrations originating at several forested areas south Ile–de–France as will be further discussed in section 5.1. These overestimated OA peaks could be related as said before to a possible overestimation of

BVOC emissions during heatwave conditions.

A strong and again sharp OA peak is simulated during the morning hours of July 13 only at the PRG site and not observed at SIRTA and RambForest (Fig. 9a–c). This peak originated in the south–western part of the Rambouillet forest and it was transported slightly north–westwards due to the weak winds (up to 2ms$^{-1}$ observed at the SIRTA site), rising in latitude. Due to a sudden change of wind direction to easterly winds, between 4 and 5 UTC in the morning (also observed at the SIRTA, see

Fig. 3c), the plume was redirected to the east, crossing only the PRG urban site and not the other sites. The discrepancy between the observed and simulated OA peak can be attributed to a number of factors, including complex meteorological conditions, the potential impact of the model spatial resolution (2 km spatial resolution) and the uncertainty in the correct placement of the OA plume direction.

Finally, two OA peaks between 30 and 40 µg m$^{-3}$ in the afternoon and night of July 19 are simulated at the three sites (Fig. 9a–c). Although the first peak is not observed by in–situ data, the second was identified and it could be attributed to the fire plume advected from the Aquitaine region in SW France (Menut et al, 2023). This event will be further described in Section 5.2. Despite these differences, the overall correlation between simulated and observed hourly OA concentrations, at the three sites is satisfactory (r between 0.62 and 0.77, Table 2).


Figure 10 presents the comparison with OC measurements from filter samples at PRG and RambForest. NMB with respect to simulations are between –3% and +4% for PRG and RambForest (under the hypothesis of a OM–to–OC ratio of 1.8 (Sciare et al., 2011)). Correlations (r) are large (0.73–0.82), indicating that averaging over 8 – 16 h smooths the short–term local





variability at the sites, thereby facilitating the following of the organic aerosols time evolution. In particular, a strong decrease
in the organic aerosol is observed at the PRG site from a concentration higher than 15 µg m$^{-3}$ to approximately 5 µg m$^{-3}$ in the
afternoon of June 18 at the end of the first heatwave and peaks of 15 – 20 µg m$^{-3}$ by the end of the second heatwave due to the
several forest fire events such that of the 19 of July, observed also at the Rambouillet Forest.

Finally, the peak observed between the 13th and 14th at PRG and not observed at RambForest site may be related to fireworks
activity in connection with the national holiday in France.


Also for secondary inorganic aerosol species, biases are much smaller for the three campaign sites than for the French
GEOD'AIR sites. For sulfate, NMB values are –1% for PRG, –12% for SIRTA, and + 37 % for RambForest (Table 2).
Especially during the first heat wave, simulated sulfate is more strongly underestimated than during the clean period (Fig. 9).
For SIRTA, NMB is –63% and –16% for the first and second heatwaves, while it is only –11% for the clean period (Table S3).
Similar tendencies are observed for the other two sites (Tables S2 and S4). These heatwave versus clean period differences are
again in agreement with the comparison with the GEOD'AIR sites in section 4.3. Again, as already made evident for the
comparison with measurements from the GEOD'AIR sites, nitrate is significantly overestimated at the three sites PRG, SIRTA
and RambForest, with NMB of +15, +21 and +148% respectively. For nitrate this large relative bias corresponds to a small
absolute one of +0.24 µg m$^{-3}$. A nitrate peak of about 15 µg m$^{-3}$ is simulated in the morning of June 16 for the three sites, and
observed to some extent (5 µg m$^{-3}$) at PRG, the only site with available measurements at this time. It is due to advection of
continental nitrate to the region under northerly wind conditions.

Figure 10 shows the BC simulations evaluation against the EC measurements from filter samples, equivalent balck carbon
(eBC) and refractory black carbon (rBC available only at RambForest) observations. Simulated BC and the measured eBC and
rBC are averaged on the filter sampling time window. The simulated BC shows a general overestimation in terms of NMB, of
~+25 % at PRG and ~+60% at RambForest (Table 3), where for the latter absolute concentrations are very small (average eBC
of 0.15 µg m$^{-3}$). Simulated concentrations at PRG larger than 1 µg m$^{-3}$ are in general not observed. For PRG, this might be
linked to the fact that this site is located on the 7th floor of a building (about 30 m a.g.l.), and then could be less sensitive to
primary emissions compared to the first model layer extending approximately to 20 m a.g.l. Simulated and observed eBC
concentrations are much lower at RambForest, mostly below 0.4 µg m$^{-3}$. The largest simulated eBC peak of 0.6 µg m$^{-3}$ during
the night of July 11 to 12 is not observed. The BC tracer analysis product (specifically developed for the ACROSS field
campaign, as detailed in Di Antonio et al., (2023a)) revealed that this peak may be caused by the advection of Paris
anthropogenic emissions to the RambForest site. Since this site is located within the forest, the discrepancy may be attributed
to the altitude at which BC concentrations are transported from the Paris agglomeration and the absence of a forest canopy
model in the simulation.

In summary, simulation to observation comparisons of PM speciation are mostly satisfying with acceptable biases for all of
the species below ±20% (except for nitrate in terms of NMB).





Although the simulation of organic aerosols is sensitive to heatwave conditions, the model demonstrates a satisfactory capacity to simulate the overall broad variability between different meteorological periods. Nevertheless, it is observed that individual
peaks lasting several hours are often not adequately simulated.

## 5. Case study illustration

In this section, we present two examples to illustrate how the WRF–CHIMERE model is able to reproduce and disentangle complex advection and chemical formation patterns of OA. This contributes to an explanation of the spatio–temporal variability encountered at the campaign measurement sites.

**5.1 Regional BSOA formation and advection during the first heatwave**

The first case study corresponds to the night of June 17 to 18 (first heatwave period), when strong OA peaks are simulated and observed. Figure 11 shows the strong BSOA build–up over forested areas within and around the Ile–de–France region during the first part of the night starting between 18 and 20 UTC and continuing until June 18 at 2 UTC. These forested areas are the Sologne region (labelled S) 130 km south of Paris, the Fontainebleau forest (F) 60 km in SSE of Paris, the Rambouillet forest
(R) 40 km in the SWW, and the Chantilly forest (C) 50 km in the NNE. For instance, at the Rambouillet forest, BSOA concentrations rise by about 20 µg m$^{-3}$ between 18 and 22 UTC corresponding to an increase rate of 5 µg m$^{-3}$ per hour. Céspedes et al. (2024) found that the atmospheric stratification was very stable during this night, suggesting stagnant conditions in the rural settings. As no advection to this site is visible in Fig. 11 during these hours, we interpret this as a BSOA formation rate at first order. OA peak at the RambForest site of about 40 µg m$^{-3}$ which is observed, although to a lesser extent (Fig. 9).
The BSOA formation is probably initiated by O$_3$ rather than NO$_3$, because simulated NO$_3$ levels are very low (about 0.5 ppt, not shown).

During the following hours, south–easterly winds advect BSOA rich air masses from source regions north–eastwards, so that they diminish at RambForest from 2 UTC onwards (Fig. 11). The BSOA rich area originating at Sologne is advected to west of the RambForest site (and thus not measured during the campaign). The one originating at Fontainebleau forest is advected
to the SIRTA and PRG sites. According to our simulations, it arrives at the SIRTA site at approximately midnight, and the maximum BSOA levels of about 30 µg m$^{-3}$ occur at 2 UTC. Unfortunately, no ACSM measurements were available at SIRTA during this night. The SIRTA site being located at the edge of the Paris agglomeration, simulated NO$_3$ concentrations are more important there and northwards of it (about 2.0 ppt, not shown) allowing probably for additional terpene oxidation and BSOA formation. Concerning the PRG site, Figure 11 shows its location at the eastern edge of the simulated BSOA plume. A small
spatial mismatch in the simulation could then explain why the plume is simulated but not observed at PRG (only enhanced OA concentrations of about 15 µg m$^{-3}$, but with no peak).

As a major conclusion, the strong night–time BSOA formation event has been successfully captured by the model. The OA peaks simulated and partly observed at the campaign sites during the night of June 17 to 18 can be traced back to night–time





BSOA build–up above forested areas within and south of Ile–de–France region and to further advection of these BSOA rich
air masses to the campaign sites.

## 5.2 Fire advection episode on the greater Paris area

The second case concerns the evening of July 19, during the second heatwave, when large OA concentrations (around 15 µg
m$^{-3}$) were measured at the three sites (Fig. S6, S7, S8). The exceptionally hot and dry conditions during summer 2022 triggered
large fire activity especially in the Landes forest in SW France leading to important forest destruction and very high local
PM$_{2.5}$ concentrations (Menut et al., 2023). The WRF–CHIMERE simulation shows that these peaks led to advection to northern
France of both primary organic aerosols from the fires and BSOA from south–western France (Aquitaine region) (Fig. 12 and
13). As stated before, in our simulation, aerosol fire emissions are taken as chemically inert, so they are by definition primary.
Several VOC species are co–emitted with fires and, in our simulation, the reactive α–pinene. These fire related BVOC
emissions are locally much stronger than those from the surrounding Landes forest, but the latter occur over a much larger
area. As a result, strong BSOA formation occurs both due to fire and forest BVOC precursors, enhanced by large ozone
concentrations for this day. During June 19, this combined fire POA and BSOA plume moves eastward (Fig. S9), passing over
the campaign sites in the evening, and impacting first the RambForest (19 UTC) and then the SIRTA and PRG (20 UTC) sites.
The model predicts for the maximum plume OA encountered during these hours that about two–thirds (~20 µg m$^{-3}$) are BSOA
(both from fire and forest BVOC) and about one–third is due to primary fire POA (~10 µg m$^{-3}$). As a conclusion, regional
advection from several hundred km upwind of primary fire and secondary OA formed from fire and forest BVOCs can
exceptionally affect the OA budget over the Ile–de–France region and can be simulated with the WRF–CHIMERE model.

## 6 Conclusions and perspectives

In this paper, we used the available measurements from the ACROSS campaign in addition to observations from air quality
and meteorological networks for a first evaluation of the coupled WRF–CHIMERE model simulation over the ACROSS
campaign in June–July 2022 in the Ile–de–France region and more widely over France. This is a required step for using the
model system to drive further in–depth analysis of different physico-chemical processes, such as for example to disentangle
chemical production, loss and transport processes of gas and aerosol species. The paper provides an overview of the
meteorological conditions over the campaign period and related major pollution (PM, O$_3$) variability over France and at the
campaign sites.
From daily maximum temperature time series and pollutant fields at the Paris urban site, three periods could be identified
during the campaign. A first heatwave period occurred from June 15 to June 18 with daily temperature maxima (T$_{max}$)
exceeding 36 °C. These elevated temperatures were partially due to heat advection by an anticyclonic pressure system situated
over Western France which draws warm air from the Saharan heat reservoir, resulting also in dust transport over the Ile–de–
France region and enhanced PM$_{10}$ surface levels over Western France for June 18.





The second heatwave period, spanning from July 12 to July 25, was characterised by a combination of very high temperatures and cooler days. During this period, southerly advection to the Ile–de–France region was again enhanced. Both heat waves were accompanied by the increased of BVOCs emissions, which led to enhanced formation of BSOA and ozone. For the intermediate period, between June 19 and July 11, ozone and PM levels were considerably lower, with air mass origins oceanic, and sometimes partially continental.


Analysis from the present work shows that the WRF model is able to satisfactorily simulate key meteorological variables and to follow the meteorological variability during the campaign, in particular during the two heat waves. Maximum daily temperature is in general simulated within ± 1°C. Maximum daily planetary boundary layer heights are often underestimated by about 200 m. The adequate simulation of temperature is important as this parameter is a general indicator for the model

ability to catch the meteorological conditions, and as in particular it governs among others the intensity of biogenic VOC emissions in the MEGAN module.

Also high and low wind speed periods and evolution of wind direction are reasonably well simulated (within ± 1 m s$^{-1}$), in particular for the Ile–de–France region. This lends confidence in the simulation of different transport regimes during the campaign period, including simulation of the direction of the pollution plume originating in the Paris region.

The WRF–CHIMERE model system generally well produces daily ozone maxima, with negative biases between –5 and 0 µg m$^{-3}$ for the air quality network sites and some larger, unexplained underpredictions up to –20 µg m$^{-3}$ during the first heatwave especially in the Ile–de–France area.

While for PM$_{10}$ the mean bias is slightly negative for the average of sites in the model domain, it is strongly positive for PM$_{2.5}$. This overestimation was unexpected, and could be related to the specific hot summer conditions, for which the model is less

well characterised. All major submicron aerosol species are concerned by this overestimation, although at lower level than the PM$_{2.5}$. The biases concerning the submicron aerosols are larger for the two heatwave periods, while slightly negative during the clean period. So there is a possible bias in OA sensitivity to temperature, which could be related to a corresponding bias in biogenic VOC emissions, or within the VBS organic aerosol scheme. It is generally observed that there is an increase in BVOCs during heat events, with a subsequent decrease occurring as a result of the hydric stress. In the current version used,

the MEGAN model does not explicitly take into account this process, which could result in an overestimation of BVOC emissions during such cases.

On the contrary, the evaluation during the campaign period at the three ACROSS campaign sites within Ile–de–France shows no systematic over–estimation of OA (biases below about ±20%), wit again larger positive biases during the heatwave periods. Inorganic aerosol and especially nitrate remains overestimated for reasons that remain to be elucidated. Largest positive biases

appear at the RambForest site, where concentrations were very small (e.g. 0.38 µg m$^{-3}$ for ammonium, 0.16 µg m$^{-3}$ for nitrate and 0.14 µg m$^{-3}$ for BC).





This result allows the model simulation to be used to explain the aerosols variability and support the scientific interpretation of the observations at the ground–based sites during the ACROSS campaign. We perform here a first model based analysis for
the two major heatwave days June 18 and July 19: major OA peaks simulated and partly observed at the campaign sites during the night of June 17 to 18 can be attributed to night–time BSOA build–up above forested areas within and south of Ile–de–France and to further advection of these BSOA–rich air masses to the campaign sites. Concerning July 19, we make evident regional advection of biomass burning aerosol and BSOA from the Aquitaine region in South West of France.

As short–term perspectives, further evaluation of the WRF–CHIMERE model system is necessary, and will be performed continuously as new campaign data will become available. Especially for biogenic VOC, such data will be useful to evaluate biogenic VOC under diversified meteorological conditions. Radical measurements (OH, $NO_3$) will be used to evaluate the model's oxidant capacity. For instance, this will enable studies affecting organic aerosol formation via different pathways, including the differentiation of precursors and oxidants. This will enable the elucidation of the interplay between anthropogenic
and biogenic sources. Future research will also concern the usage of the WRF–CHIMERE model to estimate the aerosol radiative effects, with a particular focus on the role of the anthropogenic–biogenic mixing. In order to address this goal, the next stage of this research will focus on the evaluation of the aerosol size distribution, spectral optical properties and radiation.

**Data availability.**

Datasets used in the present study from the ACROSS field campaign for the PRG and RambForest sites are available or will
be made soon available on the AERIS datacenter (https://across.aeris−data.fr/catalogue/) and DOI referenced. Some datasets already available are: the black carbon Paris–to–regional ratio (Di Antonio et al., 2023a); equivalent black carbon at the PRG site (Di Antonio et al., 2023b); $PM_1$, $PM_{2.5}$ and $PM_{10}$ (Di Antonio et al., 2023c); the refractory black carbon concentration at the RambForest site (Yu and Formenti, 2023); the non–refractory aerosol composition below and above the canopy at the RambForest site (Ferreira de Brito et al., 2023b, a); the mixing layer height at SIRTA (Kotthaus et al., 2023).
The SIRTA observatory data can be downloaded at https://sirta.ipsl.fr/ (last access: 7 April 2024).
The GEOD'AIR data are available at https://www.geodair.fr/ (last access: 7 April 2024). EEA data are available at https://discomap.eea.europa.eu/map/fme/AirQualityExport.html (last access: 24 April 2024). The MIDAS (Met Office, 2012) data are available at https://catalogue.ceda.ac.uk/uuid/220a65615218d5c9cc9e4785a3234bd0 (last access: 7 April 2024).
The HYSPLIT simulations are available on the AERIS website (Siour and Di Antonio, 2023).

**Author contributions.**

LDA and MB designed the study and discussed the results. LDA performed the full data analysis. MB and CDB supervised all the analysis work. VM and CC coordinated the ACROSS field campaign. LDA run the model simulations with contribution



from GS and MB. LDA, VM, CC, AB, ABe, MC, SC, MCi, JFB, PF, CG, OG, AG, VG, MH, LH, SK, GN, DP, JEP, EDP, VR, CY, JFD, and CDB contributed to the campaign set–up, deployment, calibration and operation of the instrumentation, and

the data collection from the PRG, SIRTA and RambForest sites. LDA, MB, and CDB wrote the paper with contributions from all co–authors.

**Competing interests.**

At least one of the (co-)authors is a member of the editorial board of Atmospheric Chemistry and Physics. Furthermore, one of the co–authors (CC) is guest editor of the Special Issue "Atmospheric Chemistry of the Suburban Forest – multiplatform

observational campaign of the chemistry and physics of mixed urban and biogenic emissions". The authors have no other competing interests to declare.

**Acknowledgments.**

The EEA is acknowledged for the air quality data used for the simulation validation. The NCEP is acknowledged for the meteorological input data used in the WRF meteorological model. The GEOD'AIR database is acknowledged for the aerosol

composition data used to validate the model simulation. Useful discussions with Marc Mallet, Yevgeny Derimian and Jean-Christophe Raut are gratefully acknowledged. We thank S. Alage, E. AlMarj, T. Bertin, P. Coll, A. Feron, M. Feingesicht, O. Guillemant, S. Harb, J. Heuser, B. Language, O. Lauret, C. Macias, F. Maisonneuve, B. Piquet–Varrault, R. Torres, P Zapf, A. Albinet, J. Allan, R. Aujay Plouzeau, M. Cayet, M. Des Forges, D. De Haan, X. Dignum, O. Favez, D. Pronovost, S. Riley, A. Rose, X. Roundtree, and M. Stepanovic for their contribution to the ACROSS field campaign deployment and filter

sampling.

**Funding.**

This work has been supported by the ACROSS and RI–URBANS projects. The ACROSS project has received funding from the French National Research Agency (ANR) under the investment program integrated into France 2030, with the reference ANR–17–MPGA–0002, and it was supported by the French National program LEFE (Les Enveloppes Fluides et

l'Environnement) of the CNRS/INSU (Centre National de la Recherche Scientifique/Institut National des Sciences de L'Univers). The RI–URBANS project has received funding from the European Union's Horizon 2020 research and innovation program under grant agreement no. 101036245. This work has been also supported by the Université Paris Est Créteil. Chenjie Yu would like to acknowledge the Marie Skłodowska–Curie COFUND postdoctoral fellowship programme supported by the Paris Region. IMT Nord Europe acknowledges financial support from the Labex CaPPA project, which is funded by the French

National Research Agency (ANR) through the PIA (Programme d'Investissement d'Avenir) under contract ANR–11–LABX–





0005–01. This project was provided with computer and storage resources by GENCI at TGCC thanks to the grant 2022–A0130107232 on the supercomputer Joliot Curie's the SKL partition. L. Hawkins and E. D. Pronovost were funded by NSF IRES 1825094.

**Special issue statement.**

This article is part of the special issue "Atmospheric Chemistry of the Suburban Forest – multiplatform observational campaign of the chemistry and physics of mixed urban and biogenic emissions". It is not associated with a conference.

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



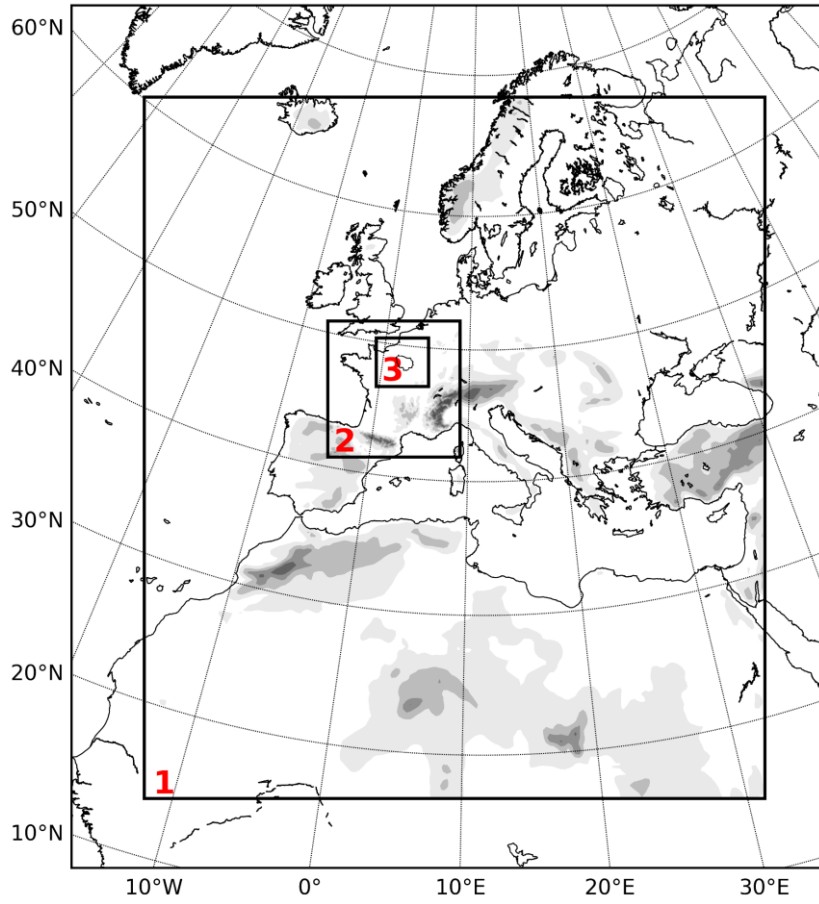

**Figure 1: Map of the three nested domains configured for the WRF–CHIMERE simulation during the ACROSS campaign 2022. The first domain (1) is at 30x30 km spatial resolution, the second (2) at 6x6 km spatial resolution, and the third (3) at 2x2 km spatial resolution covering an area centered over the Ile–de–France region. The boundaries of the Ile–de–France region are drawn in black within domain 3. The digital elevation model from the WRF output is shown in grayscale.**





**Figure 2: Temporal variations of observations during the ACROSS campaign for (a) particulate matter concentration (PM1, PM2.5, PM10) at the Paris–Rive Gauche (PRG) urban background site, (b) temperature and relative humidity at the SIRTA site, (c) wind speed and direction at the SIRTA site. The shaded an coloured arrows represent respectively the hourly and daily wind direction averages (d) mixing layer height (MLH) from (Kotthaus et al., 2023), during the ACROSS campaign 2022. SHI stands for "Saharan dust intrusion", while "FE" stands for "Fire episode". The arrows represent daily average winds.**





**Figure 3: Several meteorological variables and major pollutant concentrations at 12 UTC for (left column) 18 June, (middle column)**
**1 July and (right column) 19 July 2022 representative of the three main periods observed during the ACROSS campaign. Panels**
**(a)–(c) represent geopotential height and surface pressure from the ERA5 reanalysis (Hersbach et al., 2023b, a) ; panels (d)–(f)**
**represent the daily maximum O3; panels (g)–(i) represent the daily mean PM2.5; panels (l)–(n) represent the daily mean PM10 and**
**panels (o)–(q) the daily maximum temperature. The O3, PM2.5, and PM10 data are from the EEA database, while temperature data**
**are from the MIDAS database.**





**Figure 4: Daily bias obtained by comparing model output to observations from the MIDAS database, respectively for (left column) the full period, (middle left column) the first heatwave, (middle right column) the clean period and (right column) the second heatwave for the (a)–(d) max temperature, (e)–(h) mean temperature, (i)–(n) wind speed and (o)–(r) wind direction.**





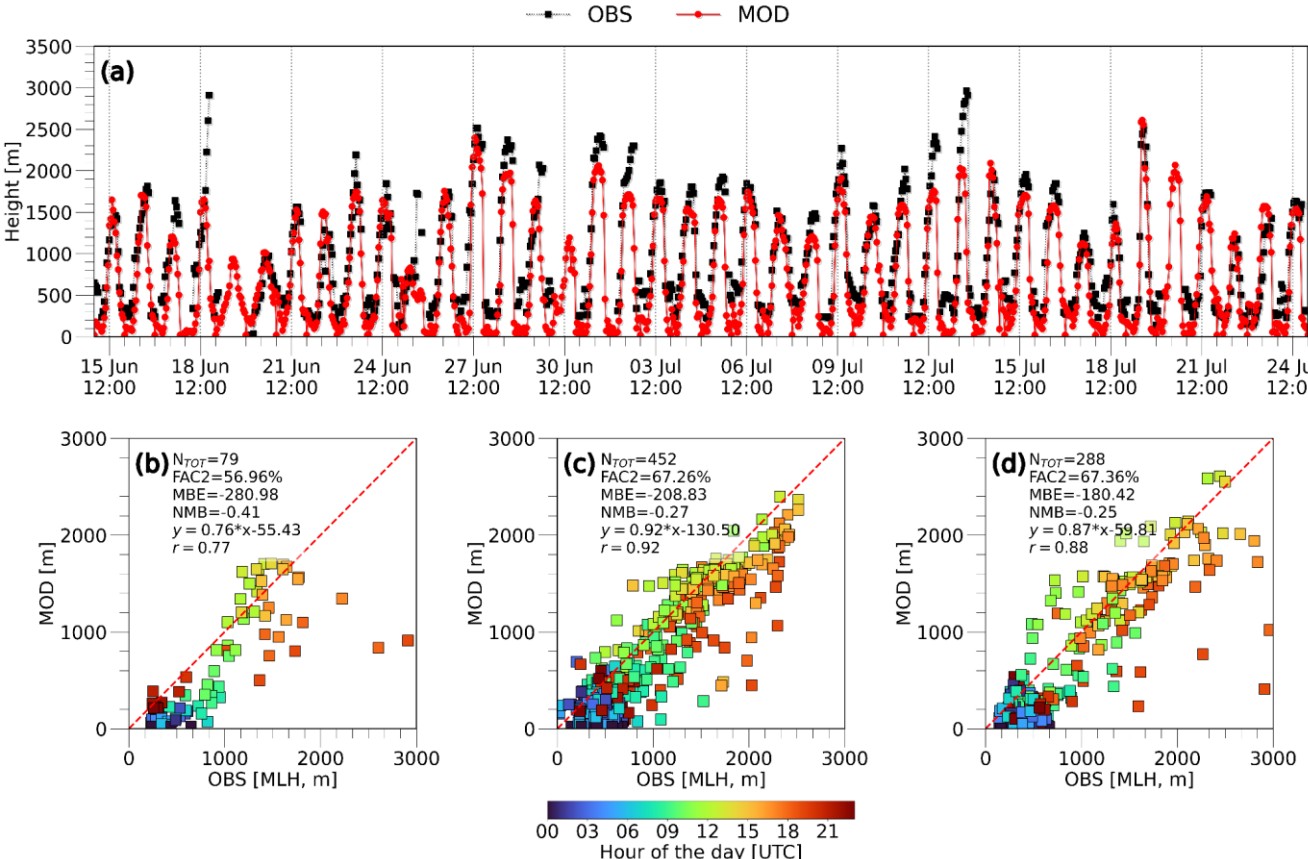

**Figure 5: (a) Comparison of the mixed layer height (MLH) time series at SIRTA between model simulations (MOD) and observations**
**(OBS); scatter plots of the modelled (PBLH) versus observed (MLH), coloured by the hour of the day for the first heatwave (b), the**
**clean period (c), the second heatwave (d).**





**Figure 6: Daily biases obtained by comparing model output to the EEA observations respectively for the full period (left column), the first heatwave (middle left column), the clean period (middle right column) and the second heatwave (right column) for the (a)–(d) O3 daily max, (e)–(h) NO2 mean, (i)–(n) PM2.5 mean and (o)–(r) PM10 mean.**



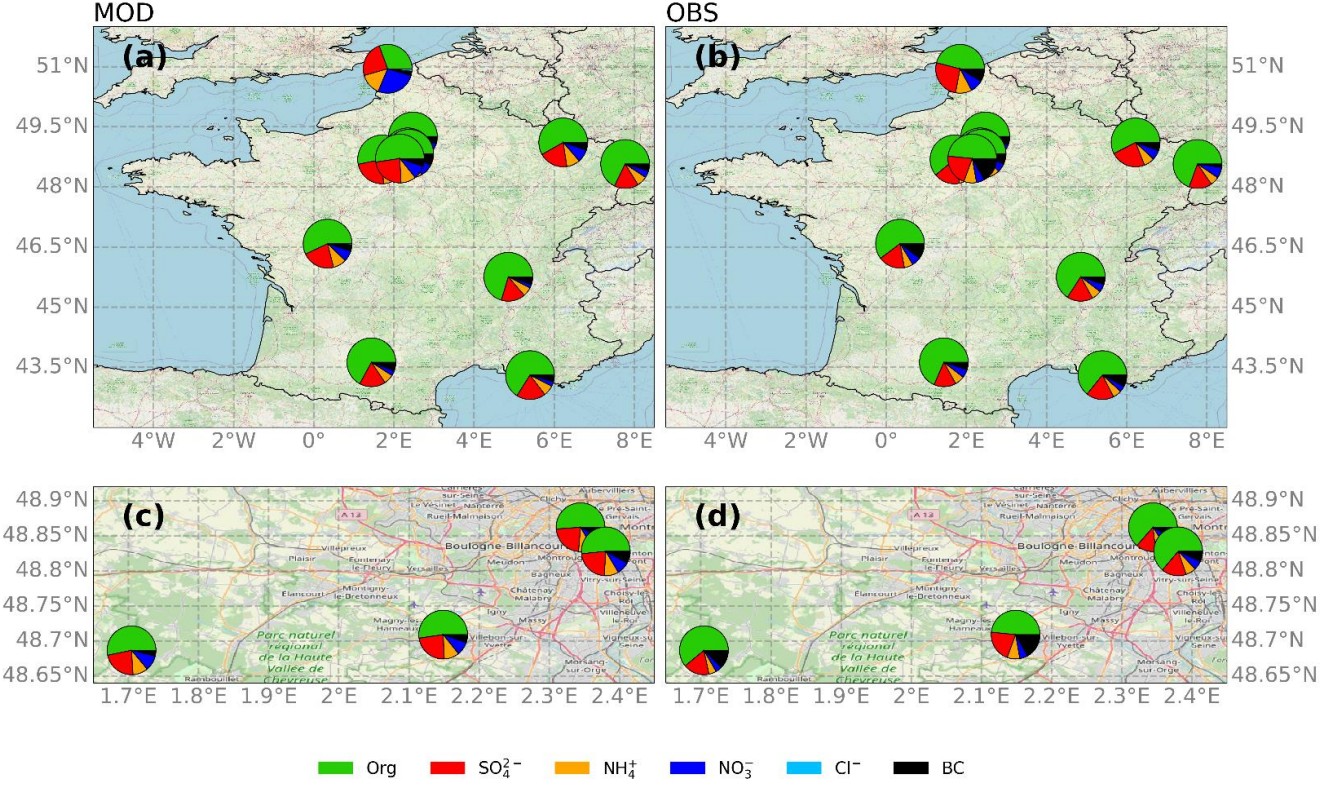

**Figure 7: Average PM1 aerosol composition from (a) the WRF–CHIMERE model (MOD), (b) the GEOD'AIR database and the PRG, SIRTA and RambForest sites (OBS). Panels (c) and (d) represent a zoom over the Ile–de–France region. Map data source: © OpenStreetMap contributors 2024. Distributed under the Open Data Commons Open Database License (ODbL) v1.0.**





**Figure 8: Comparison of simulated chemical composition to observations from the GEOD'AIR database and the PRG, SIRTA, RambForest ACROSS sites for the (a), (e), (i), (o), (s) full period, the first heatwave (panels (b), (f), (l), (p), (t)) , the clean period (panels (c), (g), (m), (q), (u)) and the second heatwave (panels (d), (h), (n), (r), (v)). Statistical metrics are calculated from data merged for all sites: Ntot, number of observations, FAC2 fraction of points within a factor of 2 limit, MBE mean bias error, NMB normalized mean bias, linear fit equation, R correlation coefficient.**



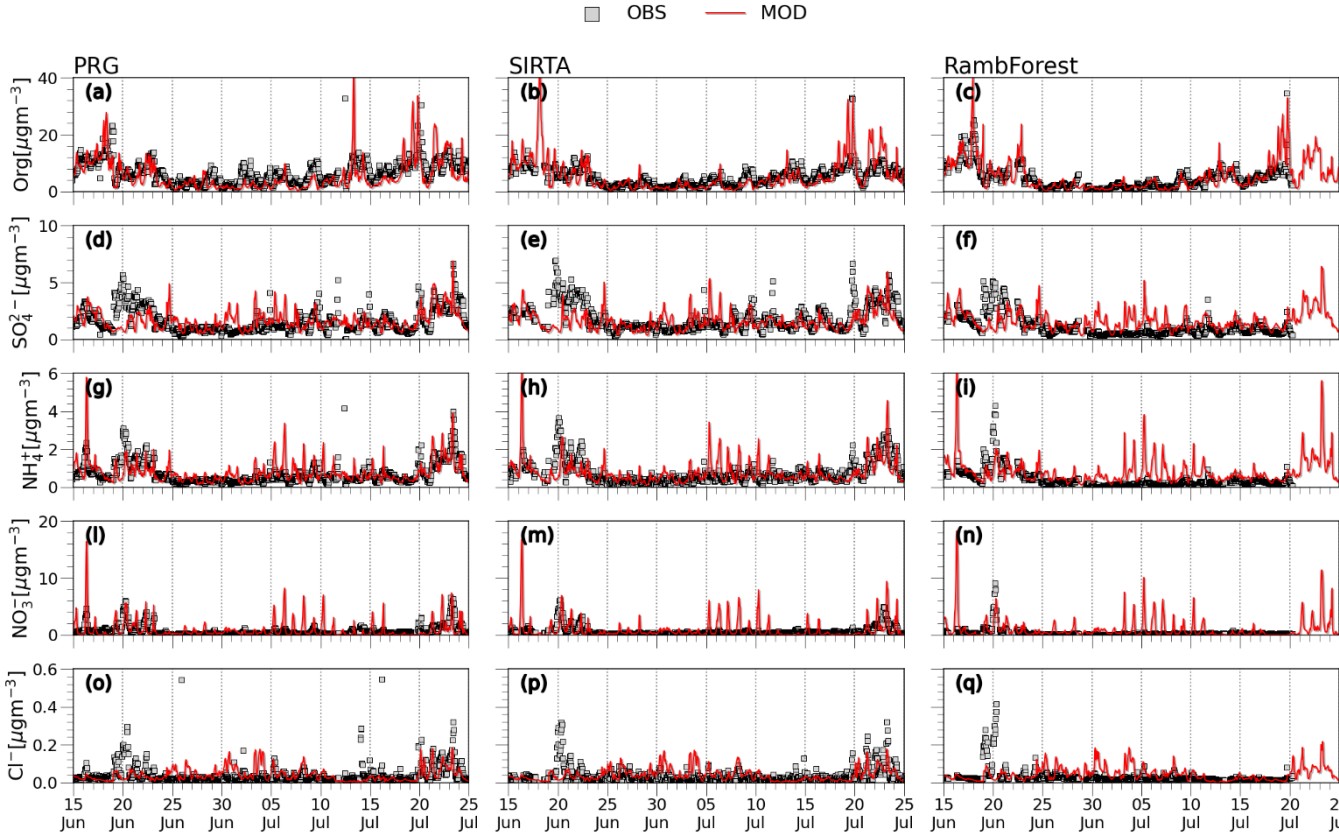

**Figure 9: Comparison of simulated chemical composition (MOD) to observations (OBS) at the PRG (ToF−ACSM model), SIRTA (ToF−ACSM model), RambForest (HR−TOF−AMS model) sites for organics, sulfate, nitrate, ammonium and chloride. For the RambForest site, only data from the AMS below the canopy are shown for plot readability. Further details on the comparison are available in Figures S6–S8 for the PRG, SIRTA and RambForest sites.**



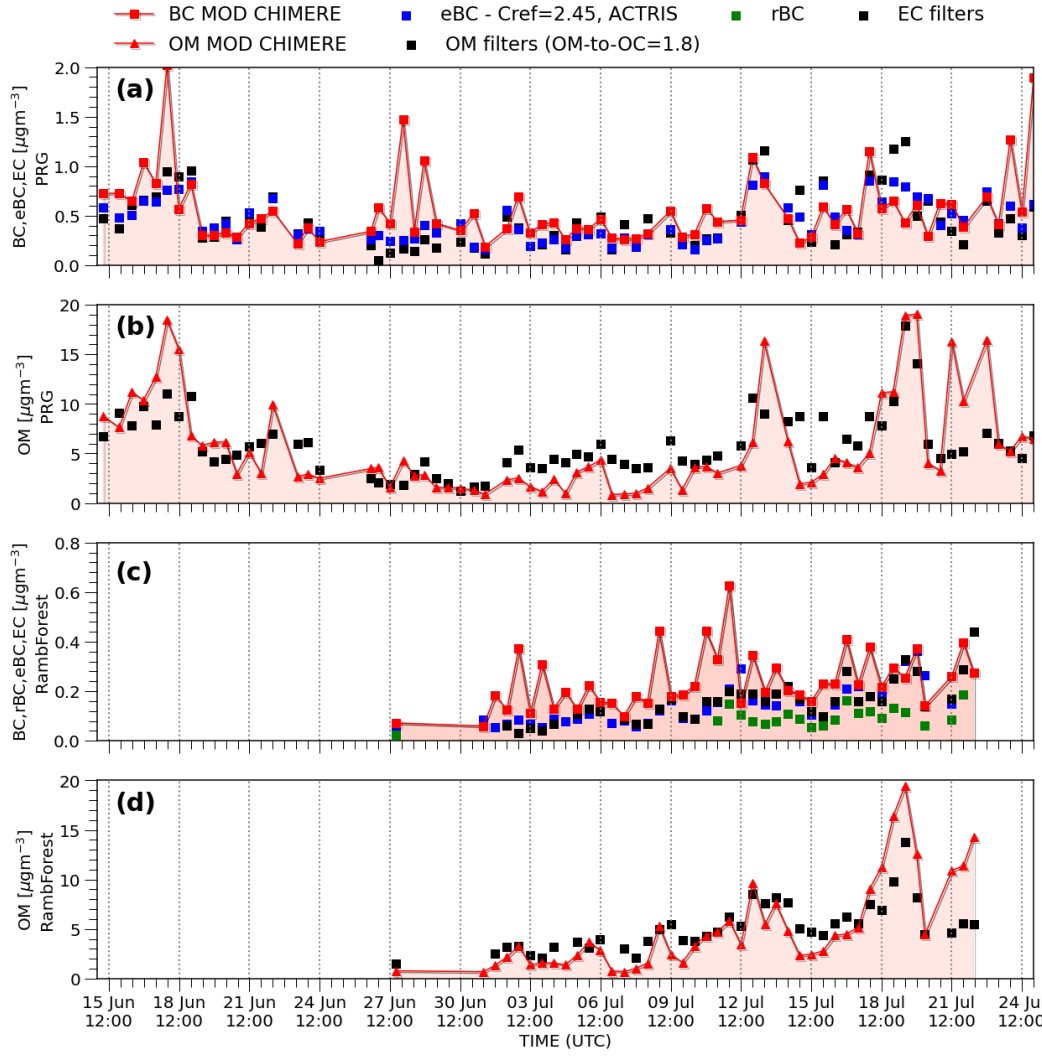

**Figure 10: Panels (a) and (c) represent the comparison of CHIMERE simulated black carbon concentration with observations at the PRG site and RambForest sites respectively: the black line represents the elemental carbon measurements (EC); the blue represents the equivalent black carbon (eBC, corrected for the multiple scattering coefficient from ACTRIS with a Cref=2.45) from AE33 measurements; the green the refractory black carbon from SP2 measurements. Panels (b) and (d) represent the comparison of CHIMERE simulated organic matter concentration with the organic matter from the filter sampling (assuming an OM–to–OC ratio of 1.8 from (Sciare et al., 2011). All the data are averaged on the filters starting and ending times.**



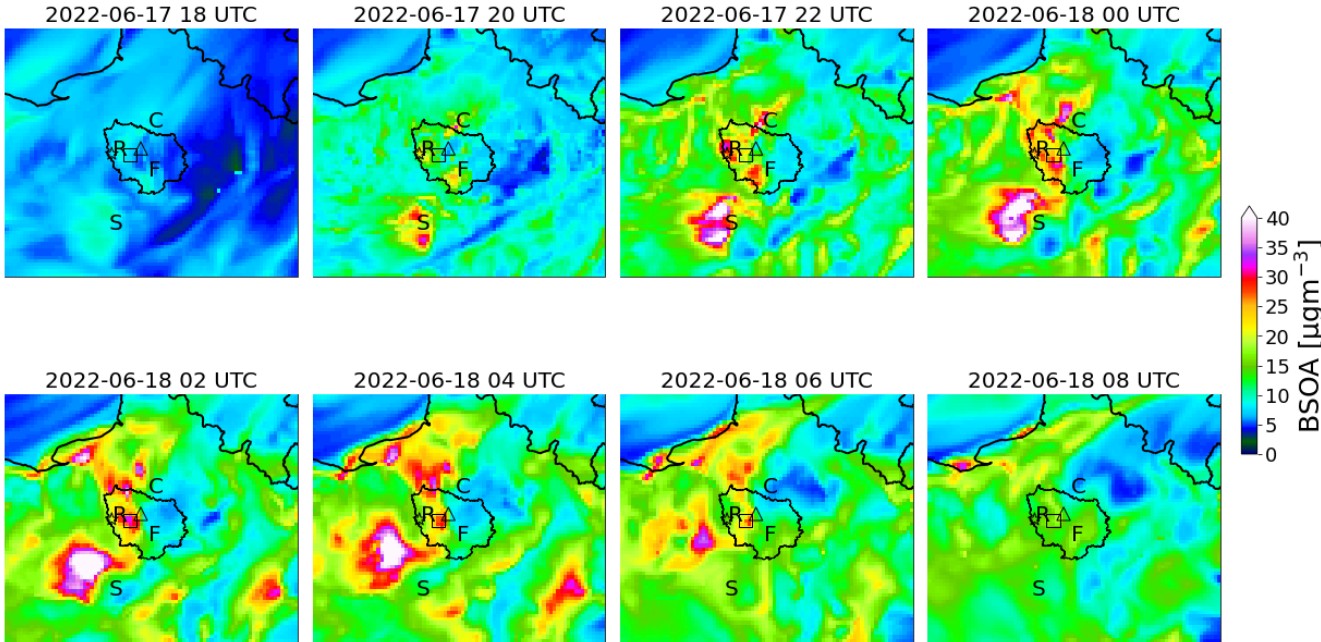

**Figure 11: Simulated biogenic secondary organic aerosol (BSOA) mass concentrations for the 17 and 18 June 2022. The letter "F" indicates the Fontainebleau forest, "R" indicates the Rambouillet forest, "S" indicates the Sologne forest and "C" indicates the Chantilly forest. The star, the square and the triangle markers indicate respectively the location of the RambForest, SIRTA and PRG sites.**



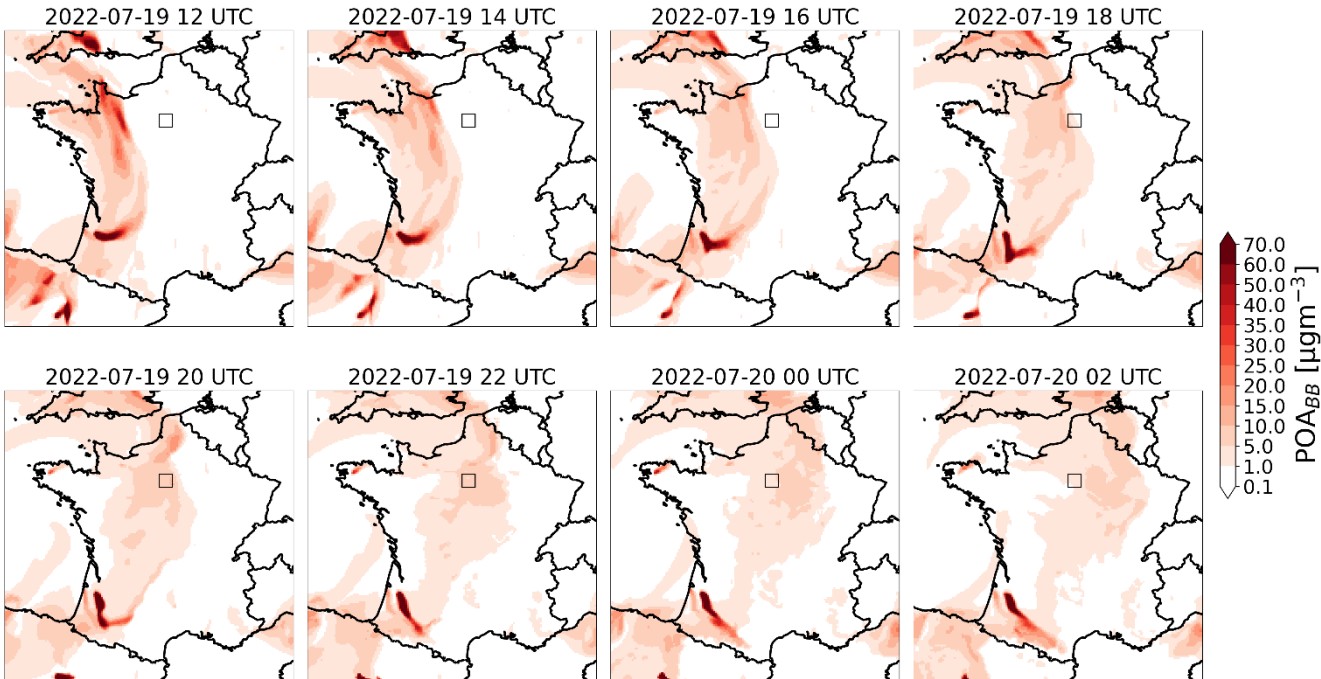

**Figure 12: Simulated primary biomass burning organic aerosol mass concentration for the fire episode of the 19 and 20 July 2022. The square marker indicates the PRG site.**





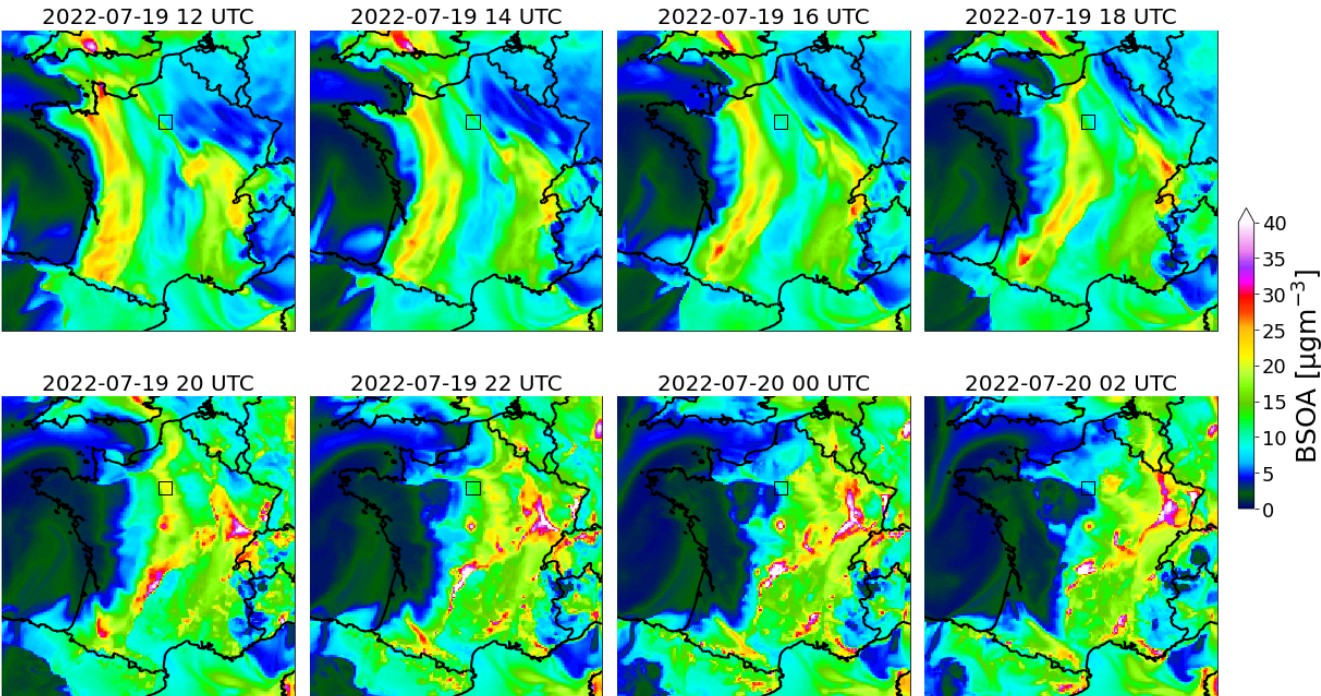

**Figure 13: Simulated biogenic secondary organic aerosol (BSOA) mass concentrations for the 19 and 20 July 2022. The square marker indicates the PRG site. Among the BSOA precursors, α–pinene stems from both fire and forest emissions.**






| Inputs | Description | Spatial resolution |
|---|---|---|
| Meteorology | WRF v. 3.7.1 model forced with the NCEP initial and boundary conditions | 1°x1° (NCEP)<br>WRF nested domains at 30, 6, 2 km |
| Initial and boundary conditions for chemistry | CAMS reanalysis (EAC4) | 0.75°x0.75° |
| Anthropogenic emissions | CAMS–GLOB–ANT v5.3 | 0.1°x0.1° |
| Biogenic emissions | Online with the MEGAN v. 2.04 model | WRF nested domains at 30, 6, 2 km |
| Fire Emissions | CAMS Global Fire Assimilation System (GFAS) | 0.1°x0.1° |
| Land use | GLOBCOVER | ~300 m |

**Table 1 Summary of the WRF–CHIMERE model inputs for the ACROSS field campaign 2022 simulation.**


| Full period (15 June – 25 July) | | | | | | | | | | | |
|---|---|---|---|---|---|---|---|---|---|---|---|
| **PRG** | | | **SIRTA** | | | **RambForest (above the canopy)** | | | **RambForest (below the canopy)** | | |
| $N_{TOT}$ | r | NMB % | $N_{TOT}$ | r | NMB % | $N_{TOT}$ | r | NMB % | $N_{TOT}$ | r | NMB % |
| Ammonium | 943 | 0.52 | 8.2 | 904 | 0.50 | 0.7 | 506 | 0.07 | 251.8 | 772 | 0.27 | 89.3 |
| Sulfate | 942 | 0.25 | −1.2 | 906 | 0.37 | −12.0 | 506 | 0.01 | 95.9 | 772 | 0.17 | 36.7 |
| Nitrate | 943 | 0.47 | 15.2 | 906 | 0.39 | 21.0 | 506 | 0.21 | 325.4 | 772 | 0.29 | 148.1 |
| Organic | 943 | 0.61 | −19.9 | 906 | 0.68 | −3.1 | 506 | 0.62 | 21.3 | 772 | 0.77 | 9.7 |
| Chloride | 943 | 0.14 | −1.7 | 888 | 0.11 | 7.9 | 506 | 0.139 | 59.4 | 772 | 0.06 | 32.8 |

**Table 2 Summary of the comparison for the PRG (urban), SIRTA (suburban) and RambForest forest (rural) sites for the aerosol refractory chemical composition measurements. Statistical metrics are: Statistical metrics are: "$N_{TOT}$" number of observations, "r" correlation coefficient, "NMB" normalized mean bias.**





| Full period (15 June – 25 July) | | | | | | |
|---|---|---|---|---|---|---|
| | PRG | | | RambForest | | |
| | $N_{TOT}$ | r | NMB % | $N_{TOT}$ | r | NMB % |
| eBC | 70 | 0.50 | 25 | 36 | 0.28 | 64 |
| EC | 70 | 0.41 | 25 | 38 | 0.38 | 60 |
| rBC | – | – | – | 21 | 0.5 | 150 |
| OM | 71 | 0.73 | –3 | 43 | 0.82 | 4 |

**Table 3 Summary of the comparison of the black carbon concentrations for the PRG and RambForest (rural) sites averaged on the filter sampling dates and times. eBC concentrations have been corrected for the ACTRIS harmonisation factor (H\*=2.45). OC has been converted to OM assuming an OM/OC ratio equal to 1.8. Statistical metrics are: "$N_{TOT}$" number of observations, "r" correlation coefficient, "NMB" normalized mean bias.**
