# Peer review of "Modelling of atmospheric variability of gas and aerosols during the ACROSS campaign 2022 in the greater Paris area: evaluation of the meteorology, dynamics and chemistry"

_EGUsphere, 2024_

## Referee Comment (RC2)

[referee-annotated manuscript omitted]

---

## Author Comment (AC1)

First, we would like to thank the reviewers for carefully reading the paper and providing valuable comments that helped to improve the quality of the manuscript. We have taken into account all the comments made by the reviewers, and have changed the paper accordingly. The details of our changes are highlighted in the text. The point-by-point answers to Reviewers #1 and #2 are provided below.

**Reviewer #1**

The manuscript presents the results of a modelling study based on the ACROSS field campaign observation over Paris and the surrounding region. A thorough evaluation of the model is accompanied by the analysis of a few specific events, focusing mainly on the origins of biogenic secondary organic aerosols (BSOA).

Let me preface the review by saying that I think that the paper is really interesting, presents important data and I'm very happy to see the combination of observations from a campaign being accompanied by modelling to help draw conclusions. However, at the current stage, the manuscript feels very technical, dealing mainly with the evaluation of the model.

The authors thank the reviewer for his/her overall positive review of the paper. Regarding the technical aspect of the paper that the reviewer regrets, this may be due to the expected role of the paper in the ACP /AMT special section "Atmospheric Chemistry of the Suburban Forest – multiplatform observational campaign of the chemistry and physics of mixed urban and biogenic emissions (ACP/AMT inter-journal SI)". The paper intends to present the WRF-CHIMERE model setup for the ACROSS field campaign in summer 2022 and a first model evaluation. This is a prerequisite for the use of the model for the interpretation of the campaign results, and it is transparent that such a model evaluation is available and referenced within the ACROSS special issue. It is then shown that the model simulations can be used to interpret particularly interesting situations of biogenic secondary organic and fire aerosol build-up and transport. The paper also gives a hopefully interesting overview of the meteorological conditions during the three campaign phases, which is also thought to be useful for this special section. At the same time, the authors acknowledge and follow the evaluator's request for a more detailed analysis of the case studies, as described below.

I have a few suggestions that I feel can increase the impact of the study and make it more in line with publications at ACP:

1. As mentioned above, the case studies feel under-represented in the manuscript and at the current stage feel more like an after-thought. I feel that this is a shame as these broaden the appeal of the study. As the manuscript is not too long I feel that the cases examined can easily be expanded to include more comprehensive analysis.

As requested by the reviewer, the two case studies are presented in a more extensive manner. Please see our detailed comments below.

2. Stemming from point 1, I feel that when it comes to the simulations, the authors stay at a surface-level interpretation and could have easily expanded the experimental design to answer questions set in the manuscript and answered in an interpretative manner. I've noted three main points that can be studied by fairly simple repeat simulations

*i. Is MEGAN overestimating emissions?* > Could be further explored by running a simulation with a biogenic emission dataset (if the possibility exists in CHIMERE) and comparing the results. Or at the very least directly comparing the MEGAN emissions to another dataset.

We thank the reviewer for this suggestion. Unfortunately, it is difficult to assess whether the MEGAN 2.1 model included in CHIMERE overestimates the biogenic emissions (isoprene, mono and sesqui-terpenes) for summer 2022. The use of other datasets for comparison is not possible as such, as these emissions are dependent on meteorology, in particular temperature and solar radiation. Incorporating alternative emission models into CTM simulations has been done in several dedicated studies (Jiang et al., 2019, Messina et al., 2016), but is beyond the scope of this paper. For a specific forest, the Landes forest in southwestern France, Cholakian et al. (2023) showed improvements in biogenic emissions by using more appropriate land use and tree species distributions. Finally, Oomen et al. (2024) adjusted European isoprene and terpene emissions. The results of these four studies, highlighting the uncertainties in the MEGAN emission model used, are added in a new Discussion section 6:

*"We first discuss here the uncertainty in model-predicted biogenic secondary organic aerosol concentrations due to uncertainty in the biogenic VOC (BVOC) emissions used in the model. BVOC emissions are predicted by the global MEGAN 2.1 module implemented in CHIMERE. While we did not find published BVOC emissions for summer 2022 from other models in the literature, several studies have compared biogenic emissions from different models and assessed the impact of the differences on secondary pollutants. For summer 2011 over Europe, Jiang et al. (2019) report 3 times higher monoterpene emissions and 3 times lower isoprene emissions over Europe with an emission model specifically developed for European tree species (named PSI model) as compared to MEGAN2.1. This leads to a factor of two increase in SOA and 7 ppb increase in ozone average concentrations over Europe in their simulations. The main differences in emissions and secondary compounds occur in the Mediterranean region, while the differences are much reduced over the northern half of France. For the Landes region, Cholakian et al. (2023), who refined land use and tree species distributions specifically for this forest, find monoterpene concentrations increased and isoprene concentrations decreased by a factor of about 2. In contrast, based on inverse modelling of TROPOMI formaldehyde columns, Oomen et al. (2024) find that initial MEGAN isoprene emissions over France were underestimated typically by a factor of 2 – 3 and monoterpene emissions by about a factor of 2 for the summers 2018 to 2021. Finally, Messina et al. (2016) compare global MEGAN2.1 isoprene emissions to those simulated by the ORCHIDEE atmosphere-vegetation interface model and find larger isoprene emissions in MEGAN over France, by a factor of about 2. In conclusion, comparisons between different BVOC emission estimates or inverse modelling show large differences typically by a factor of 2–3 with both signs. This suggests large uncertainties in emission models such as MEGAN2.1, without a clear indication of a positive or negative bias. This uncertainty in BVOC emissions is expected to have strong implications on BSOA formation (e.g. Jiang et al., 2019)."*

Jiang, J., Aksoyoglu, S., Ciarelli, G., Oikonomakis, E., El-Haddad, I., Canonaco, F., O'Dowd, C., Ovadnevaite, J., Minguillón, M. C., Baltensperger, U., and Prévôt, A. S. H.: Effects of two different biogenic emission models on modelled ozone and aerosol concentrations in Europe, Atmos. Chem. Phys., 19, 3747–3768, https://doi.org/10.5194/acp-19-3747-2019, 2019.

Cholakian, A., Beekmann, M., Siour, G., Coll, I., Cirtog, M., Ormeño, E., Flaud, P.-M., Perraudin, E., and Villenave, E.: Simulation of organic aerosol, its precursors, and related oxidants in the Landes pine forest in southwestern France: accounting for domain-specific land use and physical conditions, Atmos. Chem. Phys., 23, 3679–3706, https://doi.org/10.5194/acp-23-3679-2023, 2023.

Oomen, G.-M., Müller, J.-F., Stavrakou, T., De Smedt, I., Blumenstock, T., Kivi, R., Makarova, M., Palm, M., Röhling, A., Té, Y., Vigouroux, C., Friedrich, M. M., Frieß, U., Hendrick, F., Merlaud, A., Piters, A., Richter, A., Van Roozendael, M., and Wagner, T.: Weekly derived top-down volatile-organic-compound fluxes over Europe from TROPOMI HCHO data from 2018 to 2021, Atmos. Chem. Phys., 24, 449–474, https://doi.org/10.5194/acp-24-449-2024, 2024.

Messina, P., Lathière, J., Sindelarova, K., Vuichard, N., Granier, C., Ghattas, J., Cozic, A., and Hauglustaine, D. A.: Global biogenic volatile organic compound emissions in the ORCHIDEE and MEGAN models and sensitivity to key parameters, Atmos. Chem. Phys., 16, 14169–14202, https://doi.org/10.5194/acp-16-14169-2016, 2016.

*ii. Are biogenic BSOA emissions from forests in the region the main driver in the observed peaks?* > The authors answer this in a qualitative manner, but a simulation could be carried out by modified land use to a non-forest category to quantify the impact. (Since the authors' mention that the heatwaves conditions can be thought of as a proxy for climate change conditions, at this point even different kinds of forests could be used here to see whether changing vegetation type would mitigate this impact, but I'll agree that this starts getting out of scope!)

We thank the reviewer for this suggestion. Indeed, this is beyond the scope of this paper and could be the subject of future studies in the framework of the ACROSS field campaign. Nevertheless, below in Fig. R1-1 (also added in the supplementary material as Fig. S11), we present the time series of the different components of the simulated organic aerosol in our simulations. The figure shows the major contribution of the BSOA to the total organic aerosol at the different ACROSS sites, especially under heat wave conditions, as a proxy for future climate conditions. The OA fractions at the SIRTA periurban site and the RambForest forest site are very similar, those at the PRG urban site show slightly larger fractions for ASOA, especially an unobserved ASOA peak on July 13. However, BSOA remains by far the largest OA fraction. POA-BB, the primary fire OA, contributes episodically, on July 19 and 21. For the entire campaign, BSOA contributes 53.4%, 58.2%, 62.4% of the OA for the PRG, SIRTA and RambForest campaign sites, respectively, ASOA 29.1%, 28.5%, 26.4%, POA-BB 2.4%, 2.4%, 2.6%, POA 9.3%, 5.2%, 3.2%, and OPOA (or SI-SOA) 5.6%, 5.7%, 5.4%.

In the revised manuscript, we refer to Fig. R1-1, which has been added as new Fig. S11 in the Supplementary Material. We add the following sentences at the end of section 5.2:

"*While these two case studies demonstrate the importance of BSOA and biomass burning OA (POA-BB), the average partitioning also shows a minor fraction of ASOA (Figure S11), and makes the mixing of different OA sources over the Ile-de-France region evident. For the entire campaign, BSOA contributes to 53.4%, 58.2%, 62.4% of the OA for the PRG, SIRTA and RambForest campaign sites, respectively, ASOA 29.1%, 28.5%, 26.4%, POA-BB 2.4%, 2.4%, 2.6%, POA 9.3%, 5.2%, 3.2%, and OPOA (or SI-SOA) 5.6%, 5.7%, 5.4%.*"

[Figure]

**Figure R1-1: Simulated organic aerosol components during the ACROSS field campaign at the three different ACROSS sites. "ASOA" represents the anthropogenic organic aerosol, "BSOA" the biogenic organic aerosol, "POA" the primary organic aerosol, "OPOA" the oxidized POA (via OH), "POA-BB" the primary organic aerosol due to forest fires and "Org" the total organic aerosol.**

*iii. What is the relative importance of fire to forest BSOA emissions? >* Again, simulations can be carried out without fire emissions or without forests to further clarify results.

Removing fire emissions to isolate the relative importance of the fire and biogenic contributions to BSOA, could introduce a non-linear effect in the response to the oxidant (e.g. $O_3$) levels and aerosol concentrations in the model, and it was difficult to perform additional simulations at this stage. Instead, we compared the emission fluxes of potential BSOA precursors in the source regions from which air masses are advected to the Paris area. Figure R1-2 shows that these fire-related BVOC emissions are locally much greater than those from the surrounding Landes forest, but the latter occur over a much larger area, so that overall fire-emitted α–pinene is expected to contribute much less to BSOA formation than the forest-emitted one.

*We have added the following sentence to section 5.2:* "*These fire-related BVOC emissions are locally much greater than those from the surrounding Landes forest, but the latter occur over a much larger area, so that overall fire-emitted α–pinene is expected to contribute much less to BSOA formation than the forest-emitted one (Fig. S12).*"

[Figure]

**Figure R1-2: Daily-averaged emission fluxes of potential BSOA precursors: (left) terpenes (biogenic), and (right) α-pinene (fire) over the Gironde region (1.5°W-0.5°E, 43.8°N-45.5°N) for July 18, 2022. Fire emissions are vertically integrated.**

I believe that one of most interesting point of using models is such exploratory, hypothetical simulations that can really quantify the relative importance of different processes. At the very least, as HYSPLIT is used in the simulation, a more comprehensive analysis could be carried out, for example by looking at concentrations over the trajectories to create clearer links. Overall, I feel that there is a number of ways that the study can be expanded, but currently stays at a rudimentary level in the design and analysis.

In this paper, we use HYSPLIT v5.2.0 as a tool to visualize the origin of air masses in a qualitative way. For instance, in the first case study, these trajectories qualitatively explain the advection of dust from Northern Africa to the Paris region at higher levels (2 – 3 km above ground). However, we refrain from a more quantitative analysis and follow the pollutant concentrations along the trajectories, because trajectories induce additional uncertainties and inconsistencies with respect to the simulations, in particular because these trajectories do not represent the vertical mixing within the boundary layer. A particle model should have been used for this, but this is beyond the scope of our study. Instead, as explained above, we attempted a more quantitative analysis of model output and input in order to distinguish between different sources and to highlight the major biogenic VOC source from forests on June 18, and the mixture of fire and forest emitted biogenic VOCs over the Landes region in SW France on July 19 (which could not be separated by trajectory analysis). In fact, for both cases, the successive 2D plots shown in Figures 11 and 12 clearly indicate the origin of the air mass. Moreover, since POA-BB in our simulation is chemically inert, it has been treated as a tracer for the July 19 case study (Fig. 12).

3. The authors mention specific periods where the observed PBLH is not modelled appropriately and I think that this should be explored in more detail, as misrepresentation in the PBLH can be directly tied to errors in chemical concentrations (e.g. https://doi.org/10.5194/acp-20-2839-2020).

We thank the reviewer for this comment. In fact, accurate modelling of the atmospheric planetary boundary layer is crucial for a good reproduction of the aerosol surface concentration. The model is generally able to reproduce the diel cycle over the entire period, which was the objective of this comparison. In our analysis, we compared with the mixing layer height (MLH) product processed using the STRATfinder algorithm (Kotthaus et al., 2020). While we know that this product is not very sensitive below 230 m (making the nighttime comparison not very meaningful), we are confident in the daytime comparison, with the exception of June 18 and July 13. At this time, we know that the retrieval of the product for June 18 was difficult due to a descent of dust to the ground that made the retrieval inaccurate. In contrast, for July 13, we have no clear justification for this underestimation, as several observations show this high planetary boundary layer development. At this stage, we would tend to attribute this discrepancy to the PBL scheme used here (YSU planetary boundary layer scheme, in our study). A more detailed analysis of the atmospheric dynamics would be needed to understand whether this is a synoptic or a local condition. The latter is beyond the scope of this analysis.
* * *
Kotthaus, S., Haeffelin, M., Drouin, M.-A., Dupont, J.-C., Grimmond, S., Haefele, A., Hervo, M., Poltera, Y., and Wiegner, M.: Tailored Algorithms for the Detection of the Atmospheric Boundary Layer Height from Common Automatic Lidars and Ceilometers (ALC), Remote Sensing, 12, 3259, https://doi.org/10.3390/rs12193259, 2020.

4. I feel that a clearly-separated discussion section is missing from the manuscript as the presentation of results is mixed with interpretation through Sections 3 to 4. I have added a few comments in the attached pdf document at points that I believe could be taken out of the results' section and organised as discussion points, but I think that some overall restructuring of the manuscript around the basic idea of creating a proper discussion section would help improve the manuscript.

Following the reviewer's suggestion, we added a separate discussion section 6. Parts of the text added in response to questions from both reviewers are included into this section. The full text of this section is given below:

*"In this section, we examine some aspects of model uncertainty that we have identified in the previous section. In particular, we will discuss uncertainties in the formation of biogenic secondary organic aerosol (BSOA), related to three aspects (i) biogenic VOC (BVOC) emissions, (ii) yields of SOA formation from biogenic and anthropogenic VOC emissions, and (iii) the combined effects of (i) and (ii) in model-to-observation comparisons of OA and SOA.*

*We first discuss here the uncertainty in model-predicted biogenic secondary organic aerosol concentrations due to uncertainty in the biogenic VOC (BVOC) emissions used in the model. BVOC emissions are predicted by the global MEGAN 2.1 module implemented in CHIMERE. While we did not find published BVOC emissions for summer 2022 from other models in the literature, several studies have compared biogenic emissions from different models and assessed the impact of the differences on secondary pollutants. For summer 2011 over Europe, Jiang et al. (2019) report 3 times higher monoterpene emissions and 3 times lower isoprene emissions over Europe with an emission model specifically developed for European tree species*

*(named PSI model) as compared to MEGAN2.1. This leads to a factor of two increase in SOA and 7 ppb increase in ozone average concentrations over Europe in their simulations. The main differences in emissions and secondary compounds occur in the Mediterranean region, while the differences are much reduced over the northern half of France. For the Landes region, Cholakian et al. (2023), who refined land use and tree species distributions specifically for this forest, find monoterpene concentrations increased and isoprene concentrations decreased by a factor of about 2. In contrast, based on inverse modelling of TROPOMI formaldehyde columns, Oomen et al. (2024) find that initial MEGAN isoprene emissions over France were underestimated typically by a factor of 2 – 3 and monoterpene emissions by about a factor of 2 for the summers 2018 to 2021. Finally, Messina et al. (2016) compare global MEGAN2.1 isoprene emissions to those simulated by the ORCHIDEE atmosphere-vegetation interface model and find larger isoprene emissions in MEGAN over France, by a factor of about 2. In conclusion, comparisons between different BVOC emission estimates or inverse modelling show large differences typically by a factor of 2–3 with both signs. This suggests large uncertainties in emission models such as MEGAN2.1, without a clear indication of a positive or negative bias. This uncertainty in BVOC emissions is expected to have strong implications on BSOA formation (e.g. Jiang et al., 2019).*

*Additional uncertainty in SOA formation comes from uncertainty in the aerosol scheme itself. A recent report from (Ramboll et al., 2022), compares SOA yields for given seed OA concentrations as predicted by two-product or VBS-based SOA schemes used in various state-of-the-art models (CAMx, CHIMERE, CMAQ, GEOS-CHEM, WRF-CHEM). For instance, for a seed OA concentration of 10 $\mu$g m$^{-3}$, and under low NO$_x$ conditions, for a generic mono-terpene precursor and OH attack, the initial SOA yields, not considering further aging, range from 0.047 to 0.247, with a median of 0.182 g g$^{-1}$, the largest value being calculated with the SOA scheme used by CHIMERE in our calculation (Cholakian et al., 2018). This high yield could explain part of the BSOA overestimation in our simulations. For other precursors, and under different NO$_x$ conditions, the minimum and maximum yields typically differ by a factor of 3 to 12, and our scheme is often in the middle of the ranking. In our scheme, as in others, these yields are uniform for different mono-terpene and aromatic species, and with respect to oxidants. This is certainly a simplification, but one that is still used in recent state-of-the-art models, such as the AERO7 organic aerosol scheme used in CMAQ (Appel et al., 2021). Other models, such as the 1.5D VBS scheme implemented in CAMx (Ramboll et al., 2022), use an increased yield of BSOA species for the monoterpene + NO$_3$ reaction, which is not included in our scheme. As a consequence, our simulations may underestimate the formation of NO$_3$-initiated nocturnal BSOA. In addition, different SOA aging formulations in different schemes add additional uncertainty to the SOA evolution.*

*Considering the uncertainties in both BVOC emissions and SOA yields, we discuss here the results of previous OA simulation-observation comparisons, focusing on France. Within the Eurodelta III model intercomparison exercise, Ciarelli et al. (2019), found SOA underestimations by a factor of 2 to 10 over two Paris suburban sites for seven state-of-the-art European models. A variety of different BVOC emission inventories and SOA modules were used for this exercise including MEGAN2.1 and VBS schemes for SOA build-up (but not the VBS scheme used in the present work). On the contrary, Cholakian et al. (2023) found an average overestimation of OA (mainly BSOA) of about 60% during the LANDEX campaign in summer 2017, in the maritime pine-dominated Landes forest in southwestern France. This overestimation of BSOA occurred, even though monoterpene and isoprene precursors showed*

*good agreement after careful specification of local land use and tree distribution data. Using MEGAN2.1 BVOC emissions and a VBS scheme with aging (functionalization) for ASOA within the PMCAMX model, (Fountoukis et al., 2016) found only a small bias less than 10% in SOA (OOA OA-fraction) measurements at three urban or suburban sites in Paris. However, with a similar BVOC/SOA set-up within the WRF-CHEM model, (Barbet et al., 2016) found a factor of 6 underestimation of SOA at Puy de Dome, a mountain (at 1465 m a.s.l) background site in central France during a summer 2010 pollution episode. During the summer 2013 ChArMEx Mediterranean campaign, a CHIMERE simulation with MEGAN2.1 BVOC emissions found the best agreement in OA at Cap Corse and Mallorca (Cholakian et al., 2018), precisely with the VBS SOA scheme used in the present work, including the SOA/SVOC aging processes as functionalization, fragmentation, and formation of non-volatile SOA. A further comparison of this model setup with OA measurements at 32 European sites from the EBAS network showed an average underestimation of about 25% (Cholakian et al., 2019). To conclude this discussion, previous OA/SOA model-to-observation intercomparisons over France and Europe have shown a variety of results from strong underestimation, even with VBS-based SOA schemes, to moderate overestimation. In the light of this discussion, the observed biases in simulated OA found in our study between about ±20% during the ACROSS campaign period in June/July 2022 are moderate, even if the OA overestimations are larger (up to almost 50%) during heat wave conditions."*

Jiang, J., Aksoyoglu, S., Ciarelli, G., Oikonomakis, E., El-Haddad, I., Canonaco, F., O'Dowd, C., Ovadnevaite, J., Minguillón, M. C., Baltensperger, U., and Prévôt, A. S. H.: Effects of two different biogenic emission models on modelled ozone and aerosol concentrations in Europe, Atmos. Chem. Phys., 19, 3747–3768, https://doi.org/10.5194/acp-19-3747-2019, 2019.

Cholakian, A., Beekmann, M., Colette, A., Coll, I., Siour, G., Sciare, J., Marchand, N., Couvidat, F., Pey, J., Gros, V., Sauvage, S., Michoud, V., Sellegri, K., Colomb, A., Sartelet, K., Langley DeWitt, H., Elser, M., Prévot, A. S. H., Szidat, S., and Dulac, F.: Simulation of fine organic aerosols in the western Mediterranean area during the ChArMEx 2013 summer campaign, Atmospheric Chemistry and Physics, 18, 7287–7312, https://doi.org/10.5194/acp-18-7287-2018, 2018.

Cholakian, A., Beekmann, M., Siour, G., Coll, I., Cirtog, M., Ormeño, E., Flaud, P.-M., Perraudin, E., and Villenave, E.: Simulation of organic aerosol, its precursors, and related oxidants in the Landes pine forest in southwestern France: accounting for domain-specific land use and physical conditions, Atmos. Chem. Phys., 23, 3679–3706, https://doi.org/10.5194/acp-23-3679-2023, 2023.

Oomen, G.-M., Müller, J.-F., Stavrakou, T., De Smedt, I., Blumenstock, T., Kivi, R., Makarova, M., Palm, M., Röhling, A., Té, Y., Vigouroux, C., Friedrich, M. M., Frieß, U., Hendrick, F., Merlaud, A., Piters, A., Richter, A., Van Roozendael, M., and Wagner, T.: Weekly derived top-down volatile-organic-compound fluxes over Europe from TROPOMI HCHO data from 2018 to 2021, Atmos. Chem. Phys., 24, 449–474, https://doi.org/10.5194/acp-24-449-2024, 2024.

Messina, P., Lathière, J., Sindelarova, K., Vuichard, N., Granier, C., Ghattas, J., Cozic, A., and Hauglustaine, D. A.: Global biogenic volatile organic compound emissions in the ORCHIDEE and MEGAN models and sensitivity to key parameters, Atmos. Chem. Phys., 16, 14169–14202, https://doi.org/10.5194/acp-16-14169-2016, 2016.

Appel, K. W., Bash, J. O., Fahey, K. M., Foley, K. M., Gilliam, R. C., Hogrefe, C., ... & Wong, D. C. (2021). The Community Multiscale Air Quality (CMAQ) model versions 5.3 and 5.3. 1: system updates and evaluation. Geoscientific Model Development, 14(5), 2867-2897.

Ramboll et al : CAMx User's Guide, Version 7.20, https://www.camx.com/Files/CAMxUsersGuide_v7.20.pdf, 2022.

Ciarelli, G., Theobald, M. R., Vivanco, M. G., Beekmann, M., Aas, W., Andersson, C., Bergström, R., Manders-Groot, A., Couvidat, F., Mircea, M., Tsyro, S., Fagerli, H., Mar, K., Raffort, V., Roustan, Y., Pay, M.-T., Schaap, M., Kranenburg, R., Adani, M., Briganti, G., Cappelletti, A., D'Isidoro, M., Cuvelier, C., Cholakian, A., Bessagnet, B., Wind, P., and Colette, A.: Trends of inorganic and organic aerosols and precursor gases in Europe: insights from the EURODELTA multi-model experiment over the 1990–2010 period, Geosci. Model Dev., 12, 4923–4954, https://doi.org/10.5194/gmd-12-4923-2019, 2019.

Fountoukis, C., Megaritis, A. G., Skyllakou, K., Charalampidis, P. E., Denier van der Gon, H. A. C., Crippa, M., Prévôt, A. S. H., Fachinger, F., Wiedensohler, A., Pilinis, C., and Pandis, S. N.: Simulating the formation of carbonaceous aerosol in a European Megacity (Paris) during the MEGAPOLI summer and winter campaigns, Atmos. Chem. Phys., 16, 3727–3741, https://doi.org/10.5194/acp-16-3727-2016, 2016.

Barbet, C., Deguillaume, L., Chaumerliac, N., Leriche, M., Freney, E., Colomb, A., Sellegri, K., Patryl, L. and Armand, P. (2016). Evaluation of Aerosol Chemical Composition Simulations by the WRF-Chem Model at the Puy de Dôme Station (France). Aerosol Air Qual. Res. 16: 909-917. https://doi.org/10.4209/aaqr.2015.05.0342

I also have a few more technical questions/comments:

5. The analysis switched between the 6km and the 2km domains and it's not always clear which model results are being examined. I think that the authors should be very careful with mixing results like this. To be honest, in the current iteration of the manuscript, I don't really see the point of including the 2km domain as from what I understand results are only examined in Section 4.4. It would actually be interesting to know if the improvement in resolution is actually accompanied by an improvement in either the meteorology or chemistry. I expect that the first will be, but in my experience the latter isn't, as it's below the resolution of the anthropogenic emission dataset. However, in this case fire and biogenic emissions are very important so, if results for the 2km are improved, I think this merits some emphasis.

The evaluation for meteorology is performed over the whole of France at once and the horizontal resolution is then that of the French domain, 6 km. Indeed, we were initially interested in this overall picture of the meteorological evaluation, on a larger scale, given the advection of pollutants towards Ile-de-France.

In fact, the chemical evaluation at the higher resolution of 2 km is performed only in section 4.4 over the Ile-de-France region, but this section is crucial for the paper and we think also for the ACROSS special issue.

[Figure]

**Figure R1-3: Observed and simulated PM$_1$ aerosol chemical composition at 2 (yellow) and 6 (red) km model resolution, at the three ACROSS campaign sites.**

In order to document any improvement in the 2 km high resolution as compared to the 6 km one, we compared the two simulations at the three measurement sites (Fig. R1-3). We found only limited differences at the PRG (urban), SIRTA (suburban) and Rambouillet sites, most of the time the two simulations were undistinguishable by eye. When differences appeared, several organic aerosol and nitrate peaks were more overestimated with the 2 km high resolution run than with the 6 km run (see Fig. R1-3). However, this is a result that could not be easily anticipated. On the one hand, biogenic emissions are estimated online using the MEGAN model, based on the Leaf Area Index (LAI) and emission factors (EFs). These data are provided at the spatial resolution of approximately 1 km and then projected onto the model grid to be used by the MEGAN model. On the other hand, CAMS–GLOB–ANT v5.3 anthropogenic emissions have a resolution of 0.1° (Table 1), but are projected onto the model grid using the high resolution land use data. To avoid any misunderstanding, we have added the following sentence to section 2.1:

*"While the 6 km resolution simulations are used for comparisons with meteorological or pollutant observations over France, the finer scale 2 km resolution simulation is used for comparisons with campaign observations, especially in section 4.4. Differences between the two configurations are generally small at the three campaign sites (see Fig. S1)."*

6. Staying at the 2km domain, from what I understand from the description the vertical level structure is different between the two outer domains and the inner domain? If so this can lead to interpolation issues between the two domains and should generally be avoided. Was there a reason to do so? If they are not different then the methodology section needs some modifications to make the simulation design choices clearer.

The vertical levels do not overlap between the 6 km and 2 km resolution domains only for the CHIMERE model. The decision to use a reduced vertical level resolution is motivated by the focus on surface processes, which do not require high vertical resolution throughout the domain at high spatial resolution. In fact, although the total number of vertical layers is lower compared

to the larger domains, they are still denser near the surface (levels) where detailed resolution is most critical. Note also that the top of the lowest model layer is at 20 m above the surface in any case. Furthermore, the decision to lower the upper boundary level of the model was driven by the need to reduce computational effort. In addition, as the Ile-de-France is about 150 km from the inner domain boundary, the vertical mixing in the inner domain is likely to have time to re-equilibrate the pollutant profiles. As depicted in Fig. R1-3 above, the differences between the 6 km and 2 km resolution runs are minimal and generally not visible to the naked eye. Consequently, we conclude that any additional uncertainties associated with our model setup are likely to be small.

**7. It's common** at least in meteorological evaluation to provide RMSE values, which can better place the results for the model setup used in the context of the literature. Furthermore, I think that the study merits a comparison of the WRF-CHIMERE chemical evaluation against the literature to add some context for the reader.

As suggested, we added the figure with RMSE values for the different meteorological variables. The figure below (Fig. R1-4) has been added to the Supplementary material (Fig. S5) and commented it in Sec. 3, adding the following lines:

"*Average daily RMSE values for the above meteorological variables are also reported in Fig. S5, showing values less than 1.5 °C for daily max and mean temperature, about 1 ms-1 for the daily mean wind speed and less than 10 ° for the daily mean wind direction over the Ile-de-France region for the full campaign period.*"

The authors thank the referee for these detailed remarks. We went through all of them and made corresponding corrections in the revised paper version. We also carefully read the whole paper to avoid any spelling error, and sometimes to make sentences more clear or precise.

We also reported previous literature on model evaluation exercises in the new discussion section 6. In comparison to earlier results our study generally shows relatively small differences.

[Figure]

**Figure R1-4: Daily root mean square error (RMSE) coefficient between WRF–CHIMERE model output and observations of the MIDAS database respectively for the full period (left column), the first heatwave (middle left column), the clean period (middle right column) and the second heatwave (right column); (a)–(d) for the temperature daily max, (e)–(h) temperature daily mean, (i)–(n) wind speed daily mean and (o)–(r) wind speed daily mean.**

8. The HYSPLIT simulation configuration is completely missing from the manuscript.

The reviewer is correct, the reference to (Siour and Di Antonio, 2023) within the References section was missing. However, the HYSPLIT (v5.2.0) tool has been described within the section 2.3 as we use our WRF-CHIMERE meteorological field as input: "*In this work, we use HYSPLIT as a tool to visualize the origin of air masses in a qualitative manner, as a support the analysis of modeled pollutant fields. HYSPLIT v5.2.0 (Stein et al., 2015) back–trajectory simulations at the ACROSS ground–based sites (PRG, SIRTA and RambForest) were performed for the entire field campaign period using the WRF–CHIMERE meteorological fields as input using the domains shown in Fig. 1 (Siour and Di Antonio, 2023). The back–trajectories have a time resolution of ten minutes, which represents the exchange time between the WRF and CHIMERE models due to the coupling. The procedure of calculation is described in detail in https://across.aeris-data.fr/catalogue (last access: January 3, 2025).*"

We therefore updated the link pointing to the AERIS website and the reference where further details on the model configuration can be found:

Siour, G. & Di Antonio, L. (2023). ACROSS_LISA_WRF-CHIMERE_HYSPLIT_Backtraj_1H. [dataset]. Aeris. https://doi.org/10.25326/543

Finally, there are some minor points, language errors, typos etc. I've highlighted some in the pdf, but the manuscript merits another careful read-through by the authors.

The authors thank the referee for these detailed remarks. We went through all of them and made corresponding corrections in the revised paper version. We also carefully read the whole paper to avoid any spelling error, and sometimes to make sentences more clear or precise.

We also respond here to some of the questions asked in document:

WRF 3.7.1 is quite old by now (released on 2015). Is there any particular reason why a newer version is not used?

The choice of this specific WRF model version is driven by the fact that the CHIMERE model version employed in this study (v2020r3) is coupled only with the WRF 3.7.1 model version via the OASIS coupler. It is therefore not possible, for this CHIMERE model version, to use a newer version of WRF without developing the interface codes between the two models.

The following sentence has been added to Sec. 2.1:

"*The WRF version 3.7.1 is the one for which the coupling to CHIMERE has been performed (Menut et al., 2021). The CHIMERE v2020r3 was the most recent one when this work was started.*"

My overall recommendation would be publication after major revisions as discussed above. I hope that the authors will find the comments constructive.

Kind regards and best of luck with the revisions.

We hope we have satisfactorily addressed the referees' comments and corrections, and we are confident that these have contributed to improving the paper.

**Reviewer #2**

Di Antonio et al., presents a model evaluation study for the greater Paris area during the 2022 ACROSS campaign. The authors deployed the WRF-CHIMERE model with 3 domains configuration to probe into the model performance of organic and inorganic aerosols on top of several others meteorological parameters and gas-phase species. The model performance is presented and the authors additional described two selected cases of BSOA and wild-fires advection episodes.

The paper is generally well written (there are several typos along the manuscript, and the specificity of few sentences needs to be improved) and it is consistent with previous modelling results focusing on modelling of organic and inorganic aerosols.

Results from such comprehensive evaluation are helpful for the modelling community, and they are clearly presented thought out that paper in a well-structured manner. I have however some major comments that are needed to be addressed by the authors before I can recommend the manuscript for then final publication. I also believe that the manuscript would fit more in the journal GMD – model evaluation paper, since the authors specifically focus on the evaluation of a large array of meteorological and chemical variables.

The authors thank the referee for its overall positive reception of the paper. We agree that the paper focuses on a thorough model evaluation, especially within the frame of the summer 2022 ACROSS campaign. We are therefore convinced that the paper is a valid contribution to the ACP /AMT inter-journal special section "Atmospheric Chemistry of the Suburban Forest – multiplatform observational campaign of the chemistry and physics of mixed urban and biogenic emissions (ACROSS)". GMD is not part of this special section, and thus ACP seemed a good choice to us.

My major concern is that the description of the physical schemes lacks some keys details in the model description section. Additionally, in order to deeply probe in the overprediction of the organic and inorganic aerosol phase below the 1 micrometer aerodynamic diameter, I think the authors need to provide more details on the modeled size distribution (see my comment below).

The authors thank the referee for these remarks. Although details of the physical schemes are given in the referenced papers, and have been synthesized in the initial paper version for the sake of conciseness in a standard manner, we followed the referees request for a more detailed description within the paper, especially of the organic chemistry scheme (VBS volatility basis set). We also reported here figures and discussion with respect to the modelled size distribution. We additionally added a new discussion section revising uncertainties in the organic aerosol modelling to put our model/observation comparisons in the context of earlier exercises under similar conditions. We will indicate the added material below the referee's specific remarks.

**Comments:**

Abstract:

Line 35: "This overestimation was unexpected": Why? The authors should elaborate a bit more on that.

Line 37: "The model allows to explain how the interplay of different processes affects the fine aerosol variability and chemical composition over the campaign sites during two heatwave days: biogenic secondary organic aerosol formation in different forests around Paris, advection of wildfire aerosols, and long-range transport of Saharan dust". I suggest to describe what the

main results of the model are, rather than describing what it can be explain, i.e., the main results should be better rephrased in a concise manner in the abstract.

In order to make this part of the abstract more specific, we added or refined the abstract section as follows:

*"These differences will be confronted to existing literature, and might be increased by the hot summer 2022 conditions. For case studies during two heatwave days, the model shows the sources for two organic aerosol peaks above 20-30 µg m$^{-3}$, on one occasion, due to biogenic secondary organic aerosol formation in different forests around Paris, and on another occasion, due to advection of wildfire aerosols joint with secondary formation mainly from forest emitted BVOC's."*

We also added a paragraph in the new discussion section, to contextualize our results with respect to previous comparison studies:

*"Considering the uncertainties in both BVOC emissions and SOA yields, we discuss here the results of previous OA simulation-observation comparisons, focusing on France. Within the Eurodelta III model intercomparison exercise, Ciarelli et al. (2019), found SOA underestimations by a factor of 2 to 10 over two Paris suburban sites for seven state-of-the-art European models. A variety of different BVOC emission inventories and SOA modules were used for this exercise including MEGAN2.1 and VBS schemes for SOA build-up (but not the VBS scheme used in the present work). On the contrary, Cholakian et al. (2023) found an average overestimation of OA (mainly BSOA) of about 60% during the LANDEX campaign in summer 2017, in the maritime pine-dominated Landes forest in southwestern France. This overestimation of BSOA occurred, even though monoterpene and isoprene precursors showed good agreement after careful specification of local land use and tree distribution data. Using MEGAN2.1 BVOC emissions and a VBS scheme with aging (functionalization) for ASOA within the PMCAMX model, (Fountoukis et al., 2016) found only a small bias less than 10% in SOA (OOA OA-fraction) measurements at three urban or suburban sites in Paris. However, with a similar BVOC/SOA set-up within the WRF-CHEM model, (Barbet et al., 2016) found a factor of 6 underestimation of SOA at Puy de Dome, a mountain (at 1465 m a.s.l) background site in central France during a summer 2010 pollution episode. During the summer 2013 ChArMEx Mediterranean campaign, a CHIMERE simulation with MEGAN2.1 BVOC emissions found the best agreement in OA at Cap Corse and Mallorca (Cholakian et al., 2018), precisely with the VBS SOA scheme used in the present work, including the SOA/SVOC aging processes as functionalization, fragmentation, and formation of non-volatile SOA. A further comparison of this model setup with OA measurements at 32 European sites from the EBAS network showed an average underestimation of about 25% (Cholakian et al., 2019). To conclude this discussion, previous OA/SOA model-to-observation intercomparisons over France and Europe have shown a variety of results from strong underestimation, even with VBS-based SOA schemes, to moderate overestimation. In the light of this discussion, the observed biases in simulated OA found in our study between about ±20% during the ACROSS campaign period in June/July 2022 are moderate, even if the OA overestimations are larger (up to almost 50%) during heat wave conditions."*
* * *
Ciarelli, G., Theobald, M. R., Vivanco, M. G., Beekmann, M., Aas, W., Andersson, C., Bergström, R., Manders-Groot, A., Couvidat, F., Mircea, M., Tsyro, S., Fagerli, H., Mar, K., Raffort, V., Roustan, Y., Pay, M.-T., Schaap, M., Kranenburg, R., Adani, M., Briganti, G., Cappelletti, A., D'Isidoro, M., Cuvelier, C., Cholakian, A., Bessagnet, B., Wind, P., and Colette, A.: Trends of inorganic and organic aerosols and precursor gases in Europe: insights

from the EURODELTA multi-model experiment over the 1990–2010 period, Geosci. Model Dev., 12, 4923–4954, https://doi.org/10.5194/gmd-12-4923-2019, 2019.

Cholakian, A., Beekmann, M., Colette, A., Coll, I., Siour, G., Sciare, J., Marchand, N., Couvidat, F., Pey, J., Gros, V., Sauvage, S., Michoud, V., Sellegri, K., Colomb, A., Sartelet, K., Langley DeWitt, H., Elser, M., Prévot, A. S. H., Szidat, S., and Dulac, F.: Simulation of fine organic aerosols in the western Mediterranean area during the ChArMEx 2013 summer campaign, Atmospheric Chemistry and Physics, 18, 7287–7312, https://doi.org/10.5194/acp-18-7287-2018, 2018.

Cholakian, A., Beekmann, M., Coll, I., Ciarelli, G., and Colette, A.: Biogenic secondary organic aerosol sensitivity to organic aerosol simulation schemes in climate projections, Atmospheric Chemistry and Physics, 19, 13209–13226, https://doi.org/10.5194/acp-19-13209-2019, 2019.

Cholakian, A., Beekmann, M., Siour, G., Coll, I., Cirtog, M., Ormeño, E., Flaud, P.-M., Perraudin, E., and Villenave, E.: Simulation of organic aerosol, its precursors, and related oxidants in the Landes pine forest in southwestern France: accounting for domain-specific land use and physical conditions, Atmos. Chem. Phys., 23, 3679–3706, https://doi.org/10.5194/acp-23-3679-2023, 2023.

Fountoukis, C., Megaritis, A. G., Skyllakou, K., Charalampidis, P. E., Denier van der Gon, H. A. C., Crippa, M., Prévôt, A. S. H., Fachinger, F., Wiedensohler, A., Pilinis, C., and Pandis, S. N.: Simulating the formation of carbonaceous aerosol in a European Megacity (Paris) during the MEGAPOLI summer and winter campaigns, Atmos. Chem. Phys., 16, 3727–3741, https://doi.org/10.5194/acp-16-3727-2016, 2016.

Barbet, C., Deguillaume, L., Chaumerliac, N., Leriche, M., Freney, E., Colomb, A., Sellegri, K., Patryl, L. and Armand, P. (2016). Evaluation of Aerosol Chemical Composition Simulations by the WRF-Chem Model at the Puy de Dôme Station (France). Aerosol Air Qual. Res. 16: 909-917. https://doi.org/10.4209/aaqr.2015.05.0342

Method:

Substantial more details are needed on how the model approaches the treatment of OA (and eventually also the nitrate fraction giving the results presented here). See my comment below:

The aging reaction constants of OPOA (which I think is homogenous oxidation), ASOA and BSOA, should be all included in the text. Those parameters can highly affect the final modelled OA concentration, especially for what BSOA is concerned.

In the simulations presented in this work, a version of the VBS scheme allowing for functionalization, fragmentation and formation of non-volatile organic aerosol from semi-volatile organic compounds was activated. As the referee suspects, the aging reactions are partly homogeneous gas phase reactions (for functionalization, and for fragmentation reactions), but partly also particle phase reactions (for the formation of non-volatile aerosol). In the initial version of the paper, reference was already made to the scheme presented in (Cholakian et al., 2018), based on earlier work of (Shrivastava et al., 2013) and (Shrivastava et al., 2015). In the revised version, the aging scheme is described in much detail in the main text and the supplementary material.  The following text has been added to the paper:

"*Initial reactions rates of BSOA (mono and sesqui-terpenes, isoprene) and ASOA (aromatics, olefins, alkanes) precursors with OH, NO$_3$ and O$_3$ and the yields of semi-volatile compounds for 4 volatility bins in the 1 to 1000 µg m$^{-3}$ saturation concentration C\* range are described in Sec. S1, and Appendix H.11 of the CHIMERE model documentation (https://www.lmd.polytechnique.fr/chimere/docs/CHIMEREdoc_v2023.pdf, last access: January 3, 2025). The temperature dependent reaction rates are those used in the SAPRC07-A chemical mechanism (Carter, 2010). Yields both for high and low NO$_x$ conditions for the 4 volatility bins are those given by (Murphy and Pandis, 2009; Lane et al., 2008), and adopted for CHIMERE in (Zhang et al., 2013) and (Cholakian et al., 2018). Following this initial*

*formulation of the VBS scheme, these yields are uniform, independently of the initial oxidant attack (see discussion Sec. 6). While the yield of these semi-volatile species is temperature independent, their actual saturation concentration follows a temperature dependence given by the Clausius-Clapeyron equation with an enthalpy ΔH of 30 kJ mol$^{-1}$.*

*In the VBS scheme, formed semi-volatile species can undergo further gas phase "chemical aging reactions". In functionalization reactions, these compounds are further oxidized and acquire lower volatility (Murphy and Pandis, 2009; Lane et al., 2008). Fragmentation processes correspond to the breakup of oxidized OA compounds in the atmosphere into smaller and thus more volatile molecules. Fragmentation occurs at a 75% rate independently of the subsequent volatility (Shrivastava et al., 2015), leaving 25% to functionalization. These percentages are based on the best agreement between simulated and measured SOA as described in Shrivastava et al. (2013). Finally, non-volatile aerosol species are formed in the particle phase, as the reactions mimic oligarization reactions and formation of so-called glassy aerosol (Shrivastava et al., 2015). These processes have been included into CHIMERE by Zhang et al. (2013) (functionalization), and Cholakian et al. (2018) (fragmentation and formation of non-volatile SOA), considering one hour time scale for transformation of aerosol ASOA and BSOA species into non-volatile aerosol. They are described in Sec. S1 and Appendix H.11 of the CHIMERE documentation.*

*In the VBS scheme, primary organic aerosol is considered as semi-volatile. Volatility profiles for traffic and residential emissions for nine volatility bins according to their saturation concentration C\* ranging from 10$^{-2}$ to 10$^6$ μg m$^{-3}$ given by (Robinson et al., 2007) are implemented into CHIMERE. Semi-volatile POA species transferred into the gas phase or organic species of intermediate volatile emitted in the gas phase can undergo chemical aging reactions (functionalization, fragmentation and non-volatile species formation) as described before for ASOA and BSOA species. For POA derived species, this process is only activated for compounds having undergone three oxidation reactions (O3POA). Reactions rates and product yields are again taken from Robinson et al. (2007), Shrivastava et al. (2015), Zhang et al. (2013), and Cholakian et al. (2018). These reaction sets are given in Sec. S2 and Appendix H.6 in the documentation of the CHIMERE model. Note that POA from fire emissions have been considered as chemically inert, as uncertainties in the volatility distribution of fire emissions is large (Sinha et al., 2023)."*
* * *
Carter, W. (2010). Development of the saprc-07 chemical mechanism. Atmos Environ, 44(40):5324 –5335.

Robinson, A. L., Donahue, N. M., Shrivastava, M. K., Weitkamp, E. A., Sage, A. M., Grieshop, A. P., Lane, T. E., Pierce, J. R., and Pandis, S. N.: Rethinking Organic Aerosols: Semivolatile Emissions and Photochemical Aging, Science, 315, 1259–1262, https://doi.org/10.1126/science.1133061, 2007.

Lane, T. E., Donahue, N. M., and Pandis, S. N.: Simulating secondary organic aerosol formation using the volatility basis-set approach in a chemical transport model, Atmos. Environ., 42,7439–7451, 2008.

Murphy, B. N. and Pandis, S. N.: Simulating the formation of semivolatile primary and secondary organic aerosol in a regional chemical transport model, Environ. Sci. Technol., 43, 4722–4728, 2009.

Zhang, Q. J., Beekmann, M., Drewnick, F., Freutel, F., Schneider, J., Crippa, M., Prevot, A. S. H., Baltensperger, U., Poulain, L., Wiedensohler, A., Sciare, J., Gros, V., Borbon, A., Colomb, A., Michoud, V., Doussin, J.-F., Denier van der Gon, H. A. C., Haeffelin, M., Dupont, J.-C., Siour, G., Petetin, H., Bessagnet, B., Pandis, S. N., Hodzic, A., Sanchez, O., Honoré, C., and Perrussel, O.: Formation of organic aerosol in the Paris region during

the MEGAPOLI summer campaign: evaluation of the volatility-basis-set approach within the CHIMERE model, Atmos. Chem. Phys., 13, 5767–5790, https://doi.org/10.5194/acp-13-5767-2013, 2013.

Cholakian, A., Beekmann, M., Colette, A., Coll, I., Siour, G., Sciare, J., Marchand, N., Couvidat, F., Pey, J., Gros, V., Sauvage, S., Michoud, V., Sellegri, K., Colomb, A., Sartelet, K., Langley DeWitt, H., Elser, M., Prévot, A. S. H., Szidat, S., and Dulac, F.: Simulation of fine organic aerosols in the western Mediterranean area during the ChArMEx 2013 summer campaign, Atmospheric Chemistry and Physics, 18, 7287–7312, https://doi.org/10.5194/acp-18-7287-2018, 2018.

Shrivastava, M., Zelenyuk, A., Imre, D., Easter, R., Beranek, J., Zaveri, R. A., and Fast, J.: Implications of low volatility SOA and gas-phase fragmentation reactions on SOA loadings and their spatial and temporal evolution in the atmosphere, Journal of Geophysical Research: Atmospheres, 118, 3328–3342, https://doi.org/10.1002/jgrd.50160, 2013.

Shrivastava, M., Easter, R. C., Liu, X., Zelenyuk, A., Singh, B., Zhang, K., Ma, P.-L., Chand, D., Ghan, S., Jimenez, J. L., Zhang, Q., Fast, J., Rasch, P. J., and Tiitta, P.: Global transformation and fate of SOA: Implications of low-volatility SOA and gas-phase fragmentation reactions, Journal of Geophysical Research: Atmospheres, 120, 4169–4195, https://doi.org/10.1002/2014JD022563, 2015.

Sinha, A., George, I., Holder, A., Preston, W., Hays, M., and Grieshop, A.: Development of Volatility Distributions for Organic Matter in Biomass Burning Emissions. Environmental Science: Atmospheres. 3. 10.1039/D2EA00080F, 2023.

Additionally, the partitioning and redistribution of both organic particle and organic gases mass in the different volatility, and size bin for what particles are concerned, needs to be described in the Method sections. The authors use the VBS approach: How does the module (iteratively?) solve the partitioning equation between the total condensing material in the different volatility classes and the total pre-existing particulate mass? What it is assumed to be pre-existing particulate mass? How does the model redistribute the OA mass with respect to the volatility bins? Are all the compounds in the four volatility bins allowed to partition across all the size bins? Does the model include a kinetic approach for the resulting condensing mass fluxes (how?) as well as a kelvin effects on the vapor pressure of the different classes of organics (per size bin)? How is the growth of particles approached in the model? Those details are fundamental to understand the model results presented here and to put them in a clear prospective.

We thank the referee for these detailed questions. While the answers are detailed in the CHIMERE manual (https://www.lmd.polytechnique.fr/chimere/docs/CHIMEREdoc_v2023.pdf, last access: 5 January 2025), we will shortly synthesize them here and in the Methods section.

For partitioning and redistribution of semi-volatile organic species, we use a kinetic-thermodynamic approach. For distribution of the condensing fluxes into the different bins, we use the bulk approach from (Pandis et al. 1993) for small particles below 1.25 µm. The total flux to the bulk phase is calculated by:

$$J_i = \frac{1}{\tau_i}(G_i - G_{eq_i}),$$

where $J_i$ (µg.m−3.s−1) is the absorption or desorption flux of species i, $\tau_i$ (s) is a characteristic time of the mass transfer set to 2h, $G_i$ is the bulk gas-phase concentration of species i and $G_{eq_i}$ is the gas-phase concentration of species i at thermodynamic equilibrium. For condensation ($G_i > G_{eq_i}$), the bulk condensation flux $J_i$ is distributed into different size bins using a diffusion equation condensation given by (Seinfeld and Pandis, 1998):

$$k_i^{bin} = Number^{bin} \frac{2\pi\, D_p^{bin} D_i M_i}{RT}\, f(Kn, \alpha)$$

with Number[bin] the number of particles inside the bin, $D_p{}^{bin}$ the mean diameter of the bin, $D_i$ the diffusion coefficient for species i in air, $M_i$ its molecular weight and f(Kn, α) is a correction due to non-continuum effects and imperfect surface accommodation.

Equilibrium concentrations for the semi-volatile organic species i are related to particle concentrations $P_{eqi}$ through a temperature dependent partition coefficient $K_{pi}$ (in $m^3 \mu g^{-1}$) (Pankow, 1994):

$$G_{eqi} = \frac{P_{eqi}}{OA\, K_{pi}}$$

With OA ($\mu g\ m^{-3}$) is the preexisting organic aerosol particle mass summing over all organic compounds, ASOA, BSOA, POA, SI-SOA. In doing so, we assume ideal solubility of organic compounds, which is an approximation used in many models, because activity coefficients are difficult to determine.

The condensation and evaporation fluxes calculated with the kinetic-thermodynamic equations are solved for a chemical time step of 5 minutes, and using the iterative two step numerical solver (Menut et al., 2013; Verwer et al., 1994) with two iterations. The Kelvin effect, increasing the vapor pressure over a convex surface, is not taken into account, and would be difficult to implement into the used bulk scheme calculation. Tests showed that adding the Kelvin effect would only significantly affect the very small particles of some nanometers.

To introduce these precisions, the part of the methodological section dealing with organic aerosol has been rewritten:

*"For nucleation, the parameterization of (Kulmala et al., 1998) for sulfuric acid is used. For coagulation, we follow a formulation of coagulation kernels by (Debry et al., 2007). For condensation and evaporation, we use a kinetic-thermodynamic approach. For distribution of the condensing fluxes into the different size bins of pre-existing particles, we use the so called "bulk equilibrium approach" (Pandis et al., 1993) for small particles below 1.25 μm. The total flux to the bulk phase is calculated by:*

$$Ji = \frac{1}{\tau_i}(G_i - G_{eq_i}) \tag{1}$$

*where $J_i$ (μg $m^{-3}$ $s^{-1}$) is the absorption or desorption flux of species i, τi (s) is a characteristic time of the mass transfer set to 2h, $G_i$ is the bulk gas-phase concentration of species i and $Geq_i$ is the gas-phase concentration of species i at thermodynamic equilibrium. For condensation ($G_i > Geq_i$), the bulk condensation flux Ji is distributed into different size bins using a diffusion equation condensation given by (Seinfeld and Pandis, 1998):*

$$k_i^{bin} = Number^{bin} \frac{2\pi D_p^{bin} D_i M_i}{RT}\, f(Kn, \alpha) \tag{2}$$

*with Number[bin] the number of particles inside the bin, $D_p{}^{bin}$ the mean diameter of the bin, $D_i$ the diffusion coefficient for species i in air, $M_i$ its molecular weight and f(Kn, α) is a correction due to non-continuum effects and imperfect surface accommodation. For inorganic species ($SO_4^{2-}$, $NO_3^-$, $Cl^-$, $NH_4^+$, $Na^+$), the thermodynamic equilibria are calculated with the ISORROPIA 1 scheme (Nenes et al., 1998), for the organic semi-volatile species using Pankow et al, (1994) portioning theory considering all organic aerosol species as preexisting organic aerosol. The condensation and evaporation fluxes calculated with the kinetic-thermodynamic equation is solved for a chemical time step of 5 minutes, and using the iterative two step numerical solver (Verwer, 1994) with two iterations. The Kelvin effect, increasing the vapor pressure over a convex surface, is not taken into account, and would be difficult to implement into the used bulk*

*scheme calculation. Tests showed that adding the Kelvin effect would only significantly affect the very small particles of some nanometers.*"
* * *
Kulmala, M., A., L., and Pirjola, L. (1998). Parameterization for sulfuric acid / water nucleation rates. J. Geophys. Res., 103(No D7):8301–8307.

Debry, E., Fahey, K., Sartelet, K., Sportisse, B., and Tombette, M. (2007). Technical Note: A new SIze REsolved Aerosol Model (SIREAM). Atmos. Chem. Phys., 7:1537–1547.

Pandis, S., Wexler, A., and Seinfeld, J. (1993). Secondary organic aerosol formation and transport -ii. predicting the ambient secondary organic aerosol size distribution. Atmos. Environ., 27A:2403–2416

Seinfeld, J. H. and Pandis, S. N. (1998). Atmospheric chemistry and physics: From air pollution to climate change. Wiley-Interscience, J.Wiley, New York.

Pankow, J. F. (1994). An absorption model of gas/aerosol partition involved in the formation of secondary organic aerosol. Atmos. Environ., 28:189–193.

Verwer, J. (1994). Gauss-Seidel iteration for stiff odes from chemical kinetics. Journal on Scientific Computing, 15:1243–1250.

Menut, L., Bessagnet, B., Khvorostyanov, D., Beekmann, M., Blond, N., Colette, A., Coll, I., Curci, G., Foret, G., Hodzic, A., Mailler, S., Meleux, F., Monge, J.-L., Pison, I., Siour, G., Turquety, S., Valari, M., Vautard, R., and Vivanco, M. G.: CHIMERE 2013: a model for regional atmospheric composition modelling, Geosci. Model Dev., 6, 981–1028, https://doi.org/10.5194/gmd-6-981-2013, 2013.

Line 139: "functionalization (transfer to lower volatility)". Please correct: functionalization (transfer of organic gases to lower volatility).

Thanks for this correction. This precision has been added.

Line 143: "The POA can be oxidized by OH to form the Oxidised POA (OPOA)". I am assuming this is considered as SOA for the comparison of model results, and it does not refer to heterogenous chemistry reactions, right? Please specify.

Thanks for this comment. For simplicity, different oxidized POA species are considered as SOA. As mentioned above, SOA is formed through functionalization and fragmentation in the gas phase, as well as through the formation of non-volatile aerosol in the particle phase.

Line 144: "Four different volatility bins in the 1 to 1000 μg m$^{-3}$ saturation concentration C* range and a non–volatile species have been used to represent the ASOA and the BSOA from VOC oxidation by OH, NO3 and O3". At which reference temperature? Also, the reaction rates and mass/molar yields of each of the ASOA and BSOA precursors with respect to each different oxidant, especially for NO3 and OH, should be reported somewhere in the text along with references, preferably in the form of a table. Again, this will help to put the results in a better perspective, and facilitate the comparison with previous and future modelling studies.

Following the referee comment we added a section in the supplementary material (Sec. S1), to give reaction rates of BSOA and ASOA precursors with OH, NO$_3$ and O$_3$ and yields of semi volatile compounds. The temperature dependent reaction rates are those used in the SAPRC07-A chemical mechanism (Carter, 2010). Yields both for high and low NO$_x$ conditions are those given in the historical VBS mechanism given by Lane et al. (2008) and Murphy and Pandis, (2009), and adopted from CHIMERE in Zhang et al. (2013) and Cholakian et al. (2018). While the yield of each formed semi-volatile species ASOA1 – ASOA4 and BSOA1 – BSOA4 is

temperature independent, their actual saturation concentration follows a temperature dependence given by the Clausius-Clapeyron equation with an enthalpy $\Delta H$ of 30 kJ mol$^{-1}$. These yields are uniform for different mono-terpene species and for different aromatic species, and with respect to oxidants. This is certainly a simplification, but which is still used in recent state of the art models, for instance in the AERO7 organic aerosol scheme used in CMAQ (Appel et al., 2021). Other models, for example, the 1.5D VBS scheme within Camx (Ramboll et al., 2022), apply an increased yield of BSOA species for the monoterpene + NO$_3$ reaction, which is not included in our scheme. As a consequence, our simulations could underestimate formation of nighttime BSOA initiated by NO$_3$. Part of these explanations are included in the revised methodological section shown before. In addition, we added the following text in the new discussion Sec. 6 in order to account for the uncertainty induced by this treatment:

"*In our scheme, as in others, these yields are uniform for different mono-terpene species and for different aromatic species, and with respect to oxidants. This is certainly a simplification, but which is still used in recent state of the art models, for instance in the AERO7 organic aerosol scheme used in CMAQ (Appel et al., 2021). Other models, for example, the 1.5D VBS scheme within CAMx (Ramboll et al., 2022), apply an increased yield of BSOA species for the monoterpene + NO$_3$ reaction, which is not included in our scheme. As a consequence, our simulations could underestimate formation of nighttime BSOA initiated by NO$_3$.*"
* * *
Carter, W. (2010). Development of the saprc-07 chemical mechanism. Atmos Environ, 44(40):5324 –5335.

Appel, K. W., Bash, J. O., Fahey, K. M., Foley, K. M., Gilliam, R. C., Hogrefe, C., ... & Wong, D. C. (2021). The Community Multiscale Air Quality (CMAQ) model versions 5.3 and 5.3. 1: system updates and evaluation. Geoscientific Model Development, 14(5), 2867-2897.

Ramboll et al : CAMx User's Guide, Version 7.20, https://www.camx.com/Files/CAMxUsersGuide_v7.20.pdf, 2022.

Lane, T. E., Donahue, N. M., and Pandis, S. N.: Simulating secondary organic aerosol formation using the volatility basis-set approach in a chemical transport model, Atmos. Environ., 42,7439–7451, 2008.

Murphy, B. N. and Pandis, S. N.: Simulating the formation of semivolatile primary and secondary organic aerosol in a regional chemical transport model, Environ. Sci. Technol., 43, 4722–4728, 2009.

Zhang, Q. J., Beekmann, M., Drewnick, F., Freutel, F., Schneider, J., Crippa, M., Prevot, A. S. H., Baltensperger, U., Poulain, L., Wiedensohler, A., Sciare, J., Gros, V., Borbon, A., Colomb, A., Michoud, V., Doussin, J.-F., Denier van der Gon, H. A. C., Haeffelin, M., Dupont, J.-C., Siour, G., Petetin, H., Bessagnet, B., Pandis, S. N., Hodzic, A., Sanchez, O., Honoré, C., and Perrussel, O.: Formation of organic aerosol in the Paris region during the MEGAPOLI summer campaign: evaluation of the volatility-basis-set approach within the CHIMERE model, Atmos. Chem. Phys., 13, 5767–5790, https://doi.org/10.5194/acp-13-5767-2013, 2013.

Cholakian, A., Beekmann, M., Colette, A., Coll, I., Siour, G., Sciare, J., Marchand, N., Couvidat, F., Pey, J., Gros, V., Sauvage, S., Michoud, V., Sellegri, K., Colomb, A., Sartelet, K., Langley DeWitt, H., Elser, M., Prévot, A. S. H., Szidat, S., and Dulac, F.: Simulation of fine organic aerosols in the western Mediterranean area during the ChArMEx 2013 summer campaign, Atmospheric Chemistry and Physics, 18, 7287–7312, https://doi.org/10.5194/acp-18-7287-2018, 2018.

Line 153:" Nucleation, coagulation, condensation and dry and wet deposition processes are also addressed within this aerosol module." This is too general. Those processes need to be clearly described, at least condensation (see my previous comment).

In the revised methodological section shown before, nucleation and coagulation have been addressed by giving a reference for the used scheme:

*"For nucleation, the parameterization of (Kulmala et al., 1998) for sulfuric acid is used. For coagulation, we follow a formulation of coagulation kernels by (Debry et al., 2007)."*

As the referee has noticed from our previous comments, condensation and absorption is treated in much more detail in the revised section.
* * *
Kulmala, M., A., L., and Pirjola, L. (1998). Parameterization for sulfuric acid / water nucleation rates. J. Geophys. Res., 103(No D7):8301–8307.

Debry, E., Fahey, K., Sartelet, K., Sportisse, B., and Tombette, M. (2007). Technical Note: A new SIze REsolved Aerosol Model (SIREAM). Atmos. Chem. Phys., 7:1537–1547.

Line 347: "As a matter of fact, sites in the Alpine regions in SE France or over the Massif Central mountains in central France show close to zero or even negative correlations, indicating that the WRF–CHIMERE model does not capture well the NO2 variability for sites affected by orography". Do the authors mean complex orography? At 6 km resolution this is probably not a surprise since much higher horizontal, and vertical, resolutions are needed to reproduce the dynamics of mountain meteorology.

Indeed, we mean complex orography here, and we added the word "*complex*" in the revised version. Reproducing the dynamics of mountain meteorology was not the aim of the paper, as the focus area is the rather flat Ile-de-France region.

Line 375:

"These overestimations are modulated by meteorological conditions. They are stronger for the organic fraction under heat waves conditions, and especially larger for peak concentrations, probably triggering excessive production of BSOA". Along with my comments on the Method section, i) how was the aging of BSOA treated and ii) how does the model redistribute the organic mass in the size bins?, i.e. excessive redistribution of the condensing mass in the small diameter sizes, might generate overprediction of OA (since the authors are comparing against ACSM data with aerodynamic diameter lens cut off of 1 micrometer)? Those details are needed to understand the overprediction of OA fraction.

We answered your questions on BSOA aging and redistribution of organic mass into the size bins already above. Even if our procedure is sound, it has inherent uncertainties which can easily explain the BSOA overestimation, especially the rate constants and yields for different aging reactions. This is explained in the following text added to the new discussion section:

*"Additional uncertainty in SOA formation comes from uncertainty in the aerosol scheme itself. A recent report from (Ramboll et al., 2022), compares SOA yields for given seed OA concentrations as predicted by two-product or VBS-based SOA schemes used in various state-of-the-art models (CAMx, CHIMERE, CMAQ, GEOS-CHEM, WRF-CHEM). For instance, for a seed OA concentration of 10 µg m-3, and under low NOx conditions, for a generic mono-terpene precursor and OH attack, the initial SOA yields, not considering further aging, range from 0.047 to 0.247, with a median of 0.182 g g-1, the largest value being calculated with the SOA scheme used by CHIMERE in our calculation (Cholakian et al., 2018). This high yield could explain part of the BSOA overestimation in our simulations. For other precursors, and under different NOx conditions, the minimum and maximum yields typically differ by a factor of 3 to 12, and our scheme is often in the middle of the ranking. In our scheme, as in others, these yields are uniform for different mono-terpene and aromatic species, and with respect to oxidants. This is certainly a simplification, but one that is still used in recent state-of-the-art*

*models, such as the AERO7 organic aerosol scheme used in CMAQ (Appel et al., 2021). Other models, such as the 1.5D VBS scheme implemented in CAMx (Ramboll et al., 2022), use an increased yield of BSOA species for the monoterpene + NO3 reaction, which is not included in our scheme. As a consequence, our simulations may underestimate the formation of NO3-initiated nocturnal BSOA. In addition, different SOA aging formulations in different schemes add additional uncertainty to the SOA evolution."*

In addition, we analyzed the simulated size distribution of BSOA (the major OA compound class, see detailed response to Reviewer 1), both for the BSOA peak on June 18, and for the whole ACROSS campaign period. We found for both time periods a mass maximum at a size between approximately 0.1 and 0.2 µm, and a strongly decreasing mass profile for larger sizes, for instance a two order lower mass for the first bin above 1.1 µm (Fig. R2-1). Thus, we think

[Figure]

**Figure R2-1: Simulated BSOA size distribution during the full ACROSS campaign period (June 17 to July 20, 2022) and for the OA peak on June 18 2022, 8 UTC during the first heatwave.**

that we are on the safe side with respect to an overestimation of ACSM OA measurements (with their cut-off size around 1 µm). Unfortunately, we have no data on the observed size distribution specifically for BSOA, but only for PM and limited to below a diameter of 1µm, as our size distribution measurements were installed at the urban PRG site behind a $PM_1$ head. Averaged over the whole campaign period, the simulated particle mass peaks again between 0.1 and 0.2 µm, while the observed one peaks between 0.2 and 0.3 µm (Fig. R2-2). This finding is consistent with the behavior exhibited by the simulated size distributions and those derived from aerosol measurements in (Menut et al., 2016). Towards a size of 1 µm, simulated mass decreases by a factor of 5, while only by a factor of two for the observed mass. As a conclusion, there is an indication that the particle and then also the BSOA particle size could be underestimated in the simulation, but the strong decrease in both simulated and observed size towards the ACSM cutting diameter of 1 µm does not make it probable that this leads to a strong overestimation of $PM_1$ –BSOA.

[Figure]

**Figure R2-2: Simulated (in red) and observed (in black) SMPS particle volume size distribution measured at the PRG (urban) site (Kammer et al., 2024) during the full ACROSS campaign period (June 17 to July 20, 2022).**
* * *
Appel, K. W., Bash, J. O., Fahey, K. M., Foley, K. M., Gilliam, R. C., Hogrefe, C., ... & Wong, D. C. (2021). The Community Multiscale Air Quality (CMAQ) model versions 5.3 and 5.3. 1: system updates and evaluation. Geoscientific Model Development, 14(5), 2867-2897.

Ramboll et al : CAMx User's Guide, Version 7.20, https://www.camx.com/Files/CAMxUsersGuide_v7.20.pdf, 2022.

Cholakian, A., Beekmann, M., Colette, A., Coll, I., Siour, G., Sciare, J., Marchand, N., Couvidat, F., Pey, J., Gros, V., Sauvage, S., Michoud, V., Sellegri, K., Colomb, A., Sartelet, K., Langley DeWitt, H., Elser, M., Prévot, A. S. H., Szidat, S., and Dulac, F.: Simulation of fine organic aerosols in the western Mediterranean area during the ChArMEx 2013 summer campaign, Atmospheric Chemistry and Physics, 18, 7287–7312, https://doi.org/10.5194/acp-18-7287-2018, 2018.

Menut, L., Siour, G., Mailler, S., Couvidat, F., and Bessagnet, B.: Observations and regional modeling of aerosol optical properties, speciation and size distribution over Northern Africa and western Europe, Atmos. Chem. Phys., 16, 12961–12982, https://doi.org/10.5194/acp-16-12961-2016, 2016.

Kammer, J., Shahin, M., D'Anna, B. & Temime-Roussel, B. (2024). ACROSS_LCE_PRG_SMPS_5 min_L2. [dataset]. Aeris. https://doi.org/10.25326/658

This might also apply to the nitrate fraction: excessive partition of HNO3 might be an issue (in that case comparison with total nitrate measurements, if available, might help understanding if the model reproduces at least the sum of the phases, i.e., excluding dilution issues), but the model might be redistributing excessive nitrate mass in the lower tail of the size distribution, which should instead be allocated in the coarse mode. Bulk approach models might lead to this kind of behavior. A quick look at the size distribution of the nitrate, and eventually of the organics, might help bringing some light on that. Also, which version of ISORROPIA is the model currently using?

To answer the referees question, we show here the size distribution for the nitrate for June 16, 8 UTC when there was simulated, and to a lesser extent also observed, a nitrate peak over the Ile-de-France region. For both periods nitrate showed a maximum of mass at very small sizes, below 0.2 μm. For larger sizes above 1 μm, the mass decreases strongly. Again no size distributed nitrate observations are available for the ACROSS campaign, but the simulated size

distribution with very small nitrate particles suggests that we are on the safe side not to miss this nitrate in the ACSM measurements.

[Figure]

**Figure R2-3: Simulated nitrate size distribution during the full ACROSS campaign period (June 17 to July 20, 2022) and for the OA peak on June 16 2022, 8 UTC during the first heatwave.**

However, still part of nitrate could appear in the coarse mode, and this could affect (decrease) the nitrate part in the fine mode, and would not be accounted for in the standard CHIMERE version. Indeed, Hodzic et al. (2006) had performed a sensitivity test in which they concluded that the addition of a heterogeneous reaction of $HNO_3$ on mineral dust would shift a considerable nitrate fraction to the coarse mode (between $0.5 – 1$ µg m$^{-3}$) during summer over France. We do not think that this mechanism is responsible for the nitrate peak overestimation as it would require concurrent dust peaks which have not been observed.

In addition, Zakoura and Pandis (2018) cite many reasons which could be responsible for nitrate overestimation, among which excessive partition of nitrate into the particle phase. Unfortunately, total gas and particle phase $HNO_3$/nitrate observations, which in addition to the ACSM measurement could have allowed us to answer this question, are not available to us. So finally, at this stage, we have no satisfactory explanation for the nitrate overestimation. We can only note that the nitrate overestimation frequently occurs during morning hours under northerly wind conditions, advecting $NO_x$/$HNO_3$ and $NH_3$ rich air masses from Northern France, the BeNeLux, and the Channel region. We added the following sentences to section 4.4:

*"Other non-observed nitrate peaks are encountered for morning hours between July 5 and 17 (Fig. 9). These peaks occur under northerly wind conditions, when $NO_x$/$HNO_3$ and $NH_3$ rich air masses from Northern France, the BeNeLux, and the Channel region are advected to the campaign sites. As total nitrate measurements are not available for this study at the campaign sites at this stage, further analysis is postponed for future work."*

Zakoura, M., Pandis, S.N., 2018. Overprediction of aerosol nitrate by chemical transport models: The role of grid resolution. Atmos. Environ. 187, 390–400. https://doi.org/10.1016/j.atmosenv.2018.05.06

Hodzic, Alma & Bessagnet, Bertrand & Vautard, Robert. (2006). A model evaluation of coarse-mode nitrate heterogeneous formation on dust particles. Atmospheric Environment. 40. 4158-4171. 10.1016/j.atmosenv.2006.02.015.

Line 390:

"Contrary to the French GEOD'AIR sites, organic aerosol is not anymore systematically overestimated by simulations at the three ACROSS sites (Fig. 9). Reasons for this behaviour are not clear and need to be further investigated". This is an interesting results. At least for the nitrate fraction, an increase in the horizontal resolution of the model have shown to yield results that are in better agreement with measurements (Zakoura and Pandis, 2018). It might be worth to spend some word on that.

We thank the reviewers for this suggestion. The results presented in the paper over the Ile-de France region were based on a simulation with a 2 km horizontal grid. In order to assess if the reduced bias in simulated nitrate over Ile-de-France was due to an increase in horizontal resolution, we compared the 6 km and 2 km horizontal resolution runs over Ile-de-France. We found only limited differences at the PRG (urban), SIRTA (sub-urban) and Rambouillet sites, if any several organic aerosol and nitrate peaks were more overestimated with the 2km high resolution run (Fig. R2-4). Zakoura and Pandis (2018) have found a large positive bias reduction for nitrate simulations over Eastern US, when reducing horizontal resolution from 36 to 4 km. They attribute this effect to "artificial mixing of NO$_x$-rich plumes from major point and area sources with the background atmosphere" in coarser grid simulations. Our comparatively much less pronounced sensitivity to model resolution is probably due to the fact that the Paris agglomeration is a large surface emission source of about 30 km diameter, similarly that Rambouillet forest also has an extension of the order of 10 km. In the new discussion section we added a discussion on the effect of horizontal resolution on the modelling results over Ile-de-France.

[Figure]

**Figure R2-4: Observed and simulated PM1 aerosol chemical composition at 2 (yellow) and 6 (red) km model resolution, at the three ACROSS campaign sites.**

Zakoura, M., Pandis, S.N., 2018. Overprediction of aerosol nitrate by chemical transport models: The role of grid resolution. Atmos. Environ. 187, 390–400. https://doi.org/10.1016/j.atmosenv.2018.05.06

Line 421:

"OM–to–OC ratio of 1.8". Was the ratio applied indistinctly to all the organic species (including all the classes of volatility)? Please specify.

In fact, we applied the OM–to–OC ratio of 1.8 (Sciare et al., 2011) to all OC measurements to make them comparable with modelled OM. This value is comparable with the OM-to-OC ratios observed during the ACROSS field campaign in Ile-de-France between 1.6 and 2.1.
* * *
Sciare, J., d'Argouges, O., Sarda-Estève, R., Gaimoz, C., Dolgorouky, C., Bonnaire, N., Favez, O., Bonsang, B., and Gros, V.: Large contribution of water-insoluble secondary organic aerosols in the region of Paris (France) during wintertime, Journal of Geophysical Research: Atmospheres, 116, https://doi.org/10.1029/2011JD015756, 2011.

Section 5.1:

It is difficult to read Fig. 11 in its current form. I would suggest improving the color scale, maybe by using a Viridis palette? Also, it would be nice to overlap the wind vectors field for each of the time-step.

We tried out to use a Viridis color palette, but found that our initial choice offered a higher level of contrast. Therefore, we would like to maintain it. Indeed, it would be informative to add the wind field, but we think the pictures in Fig. 11 are much too small to add this information. Information on the wind direction can be found at Fig. 2c which indicates that average winds in the night from June 17 to 18 came from southerly directions, which also corresponds to the displacement of the BSOA plume formed over the Sologne forest (marked with an S).

Line 473:

"We interpret this as a BSOA formation rate at first order" I believe that meteorological condition might additionally play an important role in such a step increase.

Yes indeed, we fully take into account meteorological conditions in our discussion. For more clarity we added an additional sentence: *"As no advection to this site is visible in Fig. 11 during these hours, we interpret this at first order as a BSOA formation rate, and leading to enhanced surface BSOA concentrations due to suppressed vertical mixing."*

Line 544:

"In the current version used, the MEGAN model does not explicitly take into account this process, which could result in an overestimation of BVOC emissions during such cases" While this might be true from a theoretical point of view, one has to be careful with such a statements ("overestimation of BVOC emissions"). The EBAS datasets contains BVOC concentrations at several site in Europe, including France. Those measurement could be use to evaluate, or at least to give an idea, about the level of BVOCs, concentrations, currently estimated by the model (and, partially, indirectly on BVOC emissions performance). It is well known that MEGAN historically overestimate isoprene in Europe, and especially over European boreal forests, but monoterpenes concentrations, which also have higher SOA yields compared to isoprene, was not reported to be overestimated (at least as far as I am aware).

During the ACROSS campaign, extensive VOC measurements have been performed over Ile-de-France region, but are not yet available for this work, because of discrepancies between different instruments which still need to be understood. So we have to leave this VOC evaluation for a later stage. In the revised paper version, rather than suggesting a positive bias in BVOC emissions, we have discussed differences in BVOC emission data bases. We added this text to the new discussion section:

*"We first discuss here the uncertainty in model-predicted biogenic secondary organic aerosol concentrations due to uncertainty in the biogenic VOC (BVOC) emissions used in the model. BVOC emissions are predicted by the global MEGAN 2.1 module implemented in CHIMERE. While we did not find published BVOC emissions for summer 2022 from other models in the literature, several studies have compared biogenic emissions from different models and assessed the impact of the differences on secondary pollutants. For summer 2011 over Europe, Jiang et al. (2019) report 3 times higher monoterpene emissions and 3 times lower isoprene emissions over Europe with an emission model specifically developed for European tree species (named PSI model) as compared to MEGAN2.1. This leads to a factor of two increase in SOA and 7 ppb increase in ozone average concentrations over Europe in their simulations. The main differences in emissions and secondary compounds occur in the Mediterranean region, while the differences are much reduced over the northern half of France. For the Landes region, Cholakian et al. (2023), who refined land use and tree species distributions specifically for this forest, find monoterpene concentrations increased and isoprene concentrations decreased by a factor of about 2. In contrast, based on inverse modelling of TROPOMI formaldehyde columns, Oomen et al. (2024) find that initial MEGAN isoprene emissions over France were underestimated typically by a factor of 2 – 3 and monoterpene emissions by about a factor of 2 for the summers 2018 to 2021. Finally, Messina et al. (2016) compare global MEGAN2.1 isoprene emissions to those simulated by the ORCHIDEE atmosphere-vegetation interface model and find larger isoprene emissions in MEGAN over France, by a factor of about 2. In conclusion, comparisons between different BVOC emission estimates or inverse modelling show large differences typically by a factor of 2–3 with both signs. This suggests large uncertainties in emission models such as MEGAN2.1, without a clear indication of a positive or negative bias. This uncertainty in BVOC emissions is expected to have strong implications on BSOA formation (e.g. Jiang et al., 2019)."*
* * *
Jiang, J., Aksoyoglu, S., Ciarelli, G., Oikonomakis, E., El-Haddad, I., Canonaco, F., O'Dowd, C., Ovadnevaite, J., Minguillón, M. C., Baltensperger, U., and Prévôt, A. S. H.: Effects of two different biogenic emission models on modelled ozone and aerosol concentrations in Europe, Atmos. Chem. Phys., 19, 3747–3768, https://doi.org/10.5194/acp-19-3747-2019, 2019.

Cholakian, A., Beekmann, M., Siour, G., Coll, I., Cirtog, M., Ormeño, E., Flaud, P.-M., Perraudin, E., and Villenave, E.: Simulation of organic aerosol, its precursors, and related oxidants in the Landes pine forest in southwestern France: accounting for domain-specific land use and physical conditions, Atmos. Chem. Phys., 23, 3679–3706, https://doi.org/10.5194/acp-23-3679-2023, 2023.

Oomen, G.-M., Müller, J.-F., Stavrakou, T., De Smedt, I., Blumenstock, T., Kivi, R., Makarova, M., Palm, M., Röhling, A., Té, Y., Vigouroux, C., Friedrich, M. M., Frieß, U., Hendrick, F., Merlaud, A., Piters, A., Richter, A., Van Roozendael, M., and Wagner, T.: Weekly derived top-down volatile-organic-compound fluxes over Europe from TROPOMI HCHO data from 2018 to 2021, Atmos. Chem. Phys., 24, 449–474, https://doi.org/10.5194/acp-24-449-2024, 2024.

Messina, P., Lathière, J., Sindelarova, K., Vuichard, N., Granier, C., Ghattas, J., Cozic, A., and Hauglustaine, D. A.: Global biogenic volatile organic compound emissions in the ORCHIDEE and MEGAN models and sensitivity to key parameters, Atmos. Chem. Phys., 16, 14169–14202, https://doi.org/10.5194/acp-16-14169-2016, 2016.

---

## Author Response (AR2)

**Answer to the editor's review on "Modelling of atmospheric variability of gas and aerosols during the ACROSS campaign 2022 in the greater Paris area: evaluation of the meteorology, dynamics and chemistry" by L. Di Antonio et al.**

First, we would like to thank the editor for carefully reading the answer to the reviewers and the new version of the manuscript and providing valuable comments that helped to improve the quality of the manuscript. We have taken into account all the comments made by the editor, and have changed the paper accordingly. The details of our changes are highlighted in the text. The point-by-point answers are provided below.

With regard to the comment of the reviewer on the PBLH simulations, the authors do not address in their revised version the importance of errors in the chemical concentrations raised by reviewer #1, who also provides a relevant reference. Please add a comment on this in the manuscript.

We thank the editor for this suggestion. We have added the following comment in section 3.2:

"*Daily maxima are captured (± 200 m) for the majority of days, however, for about a third of cases the simulated PBLH remains below the observed MLH by more than 200 m. An inaccurate simulation of PBLH can lead to a poor representation of surface aerosol concentrations (Du et al., 2020). This could be the case of the two very hot days with Tmax higher than 33°C (June 18 and July 13), the observed MLH at SIRTA nearly reached 3 km, while simulated PBLH are only 1600 and 1900 m, respectively. ... For this day, the underestimated simulated PBLH could contribute to the overestimated OA peak (Fig. 9a).*"

Line 36: replace 'will be' by 'are'

Corrected, thanks.

Lines 118- 120: What means 'generally small differences' please provide statistical indicators. You can add those in Figure S1. Also add in this place part of your reply to the comment on the use of a smaller number of vertical layers in the 2km domain.

Thanks for this comment. Section 2.1 has been modified to take into account the comment as follows:

"*The 30 and 6 km domains of the CHIMERE model have 15 vertical layers between the surface and 300 hPa, while 10 levels are used for the 2 km domain up to 500 hPa. Although the total number of vertical layers is lower compared to the larger domains, they are still denser near the surface where detailed resolution is most critical. Indeed, differences between the two configurations are generally small at the three campaign sites (mean bias between the 6km and 2km simulations at the three campaign sites ranging between -0.4 and 0.07 µg m$^{-3}$ for organics, between -0.04 and 0.02 µg m$^{-3}$ for sulfate, between -0.01 and 0.03 for ammonium, between 0.02 and 0.08 for nitrate and -0.002 and 0.001 µg m$^{-3}$ for chloride, see Fig. S1). While the 6 km resolution simulations are used for comparisons with meteorological or pollutant observations over France, the finer scale 2 km resolution simulation is used for comparisons with campaign observations, especially in section 4.4.*"

Line 169: 'partitioning' not 'portioning'

Corrected thanks.

Line 208: please provide link to the documentation of the CHIMERE model.

Thanks, the link to the documentation has been inserted within the text.

Line 209: replace 'is' by 'are'

Done, thanks.

Lines 352-365: add somewhere in this discussion your comment on the nighttime product of the STRATfinder algorithm.

We thank the editor for this suggestion. The description of the mixing layer height (MLH) product in Section 2.2 has been updated to include the discussion of the nighttime reliability of the STRATfinder algorithm as follows:

*"Mixed layer height (MLH) derived automatically from profile observations obtained by a network of automatic lidars and ceilometers (ALC) operated in synergy with the PANAME initiative (Kotthaus et al., 2023). At SIRTA ALC profile data (Lufft CHM15k) are processed using the STRATfinder algorithm (Kotthaus et al., 2020). However, this algorithm has limited sensitivity below 230 m, making nighttime comparisons less reliable. The data also undergo additional quality control measures, developed in the context of RI–URBANS and the ABL testbed program (https://ablh.aeris-data.fr/, last access: 3 July 2024)."*

Lines 521-524: NH3 rich conditions mean also that your aerosol is probably not very acidic and this could explain why HNO3 partitions to the aerosol phase as nitrate ion. Did you check the simulated acidity of your aerosol?

Thank you for your suggestion. We have indeed checked the acidity of our aerosol in the simulation and added a Figure for the PRG urban site in the supplementary material (see Fig. R1, Fig. S12 in the supplementary material) and observed that aerosols are in general well neutralized, in particular for the overestimated nitrate peak on June 16, 2022 and on some occasions slightly acidic. Furthermore, we found that this overestimated nitrate peak is largely driven by the transport of nitrate from a strong emission source in Northern France, rather than by local nitrate formation pathways.

[Figure]

**Figure R1: Time series of cation ($\Sigma$ NH$_4^+$ + Na$^+$) and anion ($\Sigma$ 2 SO$_4^{2-}$ + NO$_3^-$ + Cl$^-$) ions concentrations at the urban PRG site during the ACROSS period.**

In the new section: Discussion

Please add also your comment/reply on the less pronounced sensitivity of nitrate simulations to the model resolution.

We thank the editor for the suggestion. Since the new Discussion Section 6 addresses uncertainties in the formation of biogenic secondary organic aerosol (BSOA), we have incorporated this aspect into Section 4.4 as follows:

"*For these days, aerosol is well neutralized to slightly acidic (see Fig. S12). Although the differences in the nitrate concentrations due to the spatial resolution of the model simulation are small (with a mean bias between the 6km and 2km spatial resolution simulation ranging from 0.02 to 0.08 µg m–3 for nitrate at the three campaign sites), when discrepancies with observations occur, the nitrate peaks are typically more overestimated in the 2km simulation compared to the 6km simulation (see Fig. S1).*"

Line 601 remove 'will'

Removed, thanks.

Line 609 replace 'report' by 'reported'

Line 611 replace 'leads' by ' led'

Line 612 replace 'occur in' by 'are computed for'

Lines 614 and 616 , 619 replace 'find' by 'found'

Line 625 'comes from THE uncertainty'

Thanks, the above corrections have been taken into account.